

# Sedimentary insights into organic matter alteration in Arctic Alaska's saline permafrost

Fabian Seemann[1,2,3], Michael Zech[1], Maren Jenrich[1], Guido Grosse[1,2], Benjamin M. Jones[4], Claire Treat[5], Lutz Schirrmeister[1], Susanne Liebner[6,7] and Jens Strauss[1]

[1]Permafrost Research Section, Alfred Wegener Institute Helmholtz Centre for Polar and Marine Research, 14473 Potsdam, Germany
[2]Institute of Geosciences, University of Potsdam, 14467 Potsdam, Germany
[3]Institute of Geography, Technische Universität Dresden, 01069 Dresden, Germany
[4]Institute of Northern Engineering, University of Alaska Fairbanks, Fairbanks, Alaska 99775, USA
[5]Department of Agroecology, Aarhus University, 8000 Aarhus, Denmark
[6]Institute of Biochemistry and Biology, University of Potsdam, 14467 Potsdam, Germany
[7]GFZ Helmholtz Centre for Geosciences, Section Geomicrobiology, 14473 Potsdam, Germany

*Correspondence to*: Fabian Seemann (fabian.seemann@awi.de)

**Abstract.** In Arctic coastal lowland regions such as northernmost Alaska, thermokarst landscapes are often underlain by saline marine deposits, a factor frequently overlooked when assessing permafrost thaw risks. To evaluate the influence of thaw and salinity on organic matter degradation and landscape dynamics, we analyzed six sediment cores from representative landforms near Utqiaġvik (Barrow Peninsula, Alaska) using a multiproxy, carbon-focused approach, with emphasis on *n*-alkane biomarkers. Undisturbed tundra uplands contained well-preserved, organic-rich Holocene sediments (~140 cm thick) overlying brackish Lateglacial deposits, indicating the presence of saline permafrost. Thermokarst lake subsidence into these substrates led to enhanced carbon degradation, as reflected by lower *n*-alkane carbon preference index (CPI) values. While West Twin Lake talik sediments exhibited brackish porewater, East Twin Lake sediments were characterized by predominantly saline porewater, indicating the presence of a cryopeg driven by salt-induced thaw-point depression. Lagoonal environments, receiving both terrestrial and lacustrine inputs, accumulate sediments under unfrozen hypersaline conditions, presenting a high potential for organic carbon degradation. Carbon proxy signatures statistically distinguish perennially frozen uplands, unfrozen lakes, refrozen drained lake basins, and lagoonal settings. Our results demonstrate that salt-bearing deposits, as found in all investigated sites, are vulnerable to active layer deepening, talik and cryopeg formation, and shoreline erosion. These processes accelerate organic matter degradation and alter landscape trajectories. Our study underscores the need to better understand the role of saline permafrost in Arctic coastal lowlands and its broader implications under ongoing climate change.

## 1 Introduction

Arctic amplification is warming the polar north almost four times as fast as the global mean (Chylek et al., 2022; Rantanen et al., 2022). This amplification contributes to permafrost warming, increasing active layer depths, and a shrinking permafrost



extent, which relates almost linearly to climate change (Biskaborn et al., 2019; Liu et al., 2024; McGuire et al., 2016; Nitzbon et al., 2024; Smith et al., 2022). The terrestrial permafrost region stores about three times the amount of organic

carbon of global vegetation, which is vulnerable to decomposition with warming temperatures and permafrost thaw (Schuur et al., 2022; Strauss et al., 2025).

On vast Arctic coastal plains, like in northern Alaska, permafrost thaw manifests in several types of landforms, including thermokarst lakes and lagoons. Thermokarst processes are accelerating in the Alaskan tundra (Chen et al., 2021) while Nitze et al. (2017) describe a thermokarst lake drainage trend. In newly developed drained thermokarst lake basins (DLBs),

permafrost starts to re-aggrade, but as the climate continues to warm, taliks (unfrozen areas) may increasingly remain in the sediments (Farquharson et al., 2022; Jones et al., 2022).

Furthermore, saline permafrost is present under Holocene deposits on Arctic coastal plains, like in Alaska, Canada and Siberia (Brigham-Grette and Hopkins, 1995; Brouchkov, 2002, 2003; Eisner et al., 2005; Osterkamp, 1989). In these sediments, which are often deposited under past shallow marine conditions, the thawing point is depressed due to the high

salt content, leading to a higher vulnerability to ground warming or even unfrozen cryotic conditions. When unfrozen cryotic areas are present, these are referred to as cryopegs (van Everdingen, 2005). Jones et al. (2023) observed intensified permafrost degradation in saline thermokarst lake deposits, stressing the importance of organic matter dynamics in cryopegs. Along the coastlines, the combination of sea-level rise and erosion leads to significant land loss, altering the carbon cycle and with that also greenhouse gas dynamics (Creel et al., 2024; Irrgang et al., 2022; Jenrich et al., 2024, 2025a; Nielsen et

al., 2022; Vonk et al., 2025). Coastal and gully erosion can lead to thermokarst lake drainage into the ocean but may also open up lakes and turn them into new thermokarst lagoons (Arp et al., 2010; Jenrich et al., 2021; Jones et al., 2020). The importance of such transitional environments is demonstrated by the abundance of lagoons along some coasts. For example, more than 70 % of the Alaskan Beaufort Sea coastline is characterized by both thermokarst and non-thermokarst lagoons (Harris et al., 2017; Jenrich et al., 2025b).

To understand potential carbon losses due to permafrost degradation in such transitional landscapes, we need to understand sediment properties from these complex and dynamic systems characterized by regular or saline permafrost and diverse permafrost histories. On this issue, Giest et al. (2025) recently quantified a stronger degradation signal of saline deposits compared to non-salt-influenced sites. Since coastal permafrost regions can vary substantially in their salinity(Jenrich et al., 2021), more differentiated investigations are needed. Organic carbon proxies such as organic carbon to nitrogen ratios (C:N),

organic carbon isotopes ($\delta^{13}$C, $^{14}$C) and hydrocarbons ($n$-alkanes) are known to reflect organic matter origin and its degradation state, making them useful tools in tracing paleoenvironmental change and present carbon dynamics in permafrost regions (e.g., Fuchs et al., 2019; Strauss et al., 2015). Organic matter quality - in the sense of its future degradation potential - can, for example, be assessed by the $n$-alkane carbon preference index (CPI), a proxy that is increasingly applied in permafrost carbon studies (e.g., Haugk et al., 2022; Yang et al., 2023).

In this study, we investigate the transformation of organic carbon during permafrost degradation by studying core samples from a drilling transect along thaw and salinity gradients encompassing a diverse talik- and cryopeg-affected thermokarst



terrain in Arctic Alaska. Accordingly, we address the following research questions: (1) which paleoenvironmental and modern processes and conditions shape today's sediment characteristics?, (2) what is the organic matter quality in thermokarst landforms on the Barrow Peninsula?, and (3) how do landscape dynamics affect organic matter mobilization?
To achieve this, we employ a sedimentary organic carbon-centered multiproxy approach, including the aforementioned parameters.

## 2. Material and methods

### 2.1 Study area

The study area lies ca. 10 km east of Utqiaġvik (71.276369 N, 156.452961 W, Fig. 1). The Barrow Peninsula covers an area
of about 1800 km²(Lara et al., 2020). Geomorphologically, it is part of the Younger Outer Coastal Plain, which belongs to the Arctic Coastal Plain (Hinkel et al., 2005).

In the period 2016-2020, the mean annual air temperature was -7.8 °C and the mean annual precipitation was 200 mm. Both variables have been experiencing a significant increasing trend between 1981 and 2020 (Rawlins, 2021). The landscape is characterized by continuous permafrost with mean annual ground temperatures of approx. -6 °C at the permafrost table (Obu
et al., 2019). The permafrost thickness is about 400 m (Brown et al., 2003) and the active layer (late season thaw) depth in Utqiaġvik ranged between 29 cm and 47 cm in the monitoring period 1995-2019 (Nyland et al., 2021).

The ecoregion of the study area is Arctic Tundra (Lara et al., 2025). Tundra uplands (primary or remnant surfaces) are shaped by high-centered polygons. Today's vegetation communities are dependent on landscape position. High and flat centered polygons are dominated by dry and moist graminoids, sedges and dwarf shrubs. Wet and seasonally flooded
positions such as DLBs and troughs are predominantly characterized by graminoid species, sedges and *Sphagnum* mosses (Eisner et al., 2005; Lara et al., 2015; Wolter et al., 2024). The ground consists of ice- and organic-rich Holocene deposits, which are underlain by late Pleistocene sandy and silty marine sediments (Eisner et al., 2005). These saline sediments accumulated when sea levels were higher compared to today (Brigham-Grette and Hopkins, 1995; Brouchkov, 2003). The Cretaceous bedrock consists of sedimentary rock (Black, 1964).
Thermokarst lakes and DLBs cover about 22 % and 50 %, respectively (Hinkel et al., 2003; Jones et al., 2022). Both in our study investigated lakes, West Twin Lake and East Twin Lake are characterized by floating ice regimes, yet West Twin Lake is a freshwater lake while East Twin Lake exhibits brackish water (Jones et al., 2023). In DLBs, permafrost re-aggradates after lake drainage, initializing polygonal patterns while remnant ponds may remain (Andresen and Lougheed, 2015; Eisner et al., 2005; Jones et al., 2022; Ling and Zhang, 2004). The former lake of the DLB, which is investigated in this study, was
estimated to have drained about 100-150 years ago following coastal erosion (Brown et al., 2003). The erosion rate of Elson Lagoon north and east of the investigated terrestrial sites is 0.3-5.0 m per year (Gibbs and Richmond, 2017; Osterkamp and Harrison, 1985). South of the DLB a small semi-open thermokarst lagoon is fed by multiple streams (Fig. 1). Elson Lagoon itself is a large (125 km²) but shallow (< 4m) coastal lagoon with active deposition processes occurring (Zimmermann et al.,



2022). Subsea permafrost exists beneath hypersaline lagoon sediments (Osterkamp and Harrison, 1982, 1985). The Barrow

spit as well as barrier islands mark the transition to the Alaskan Beaufort Sea (Brown et al., 2003; Zimmermann et al., 2022).

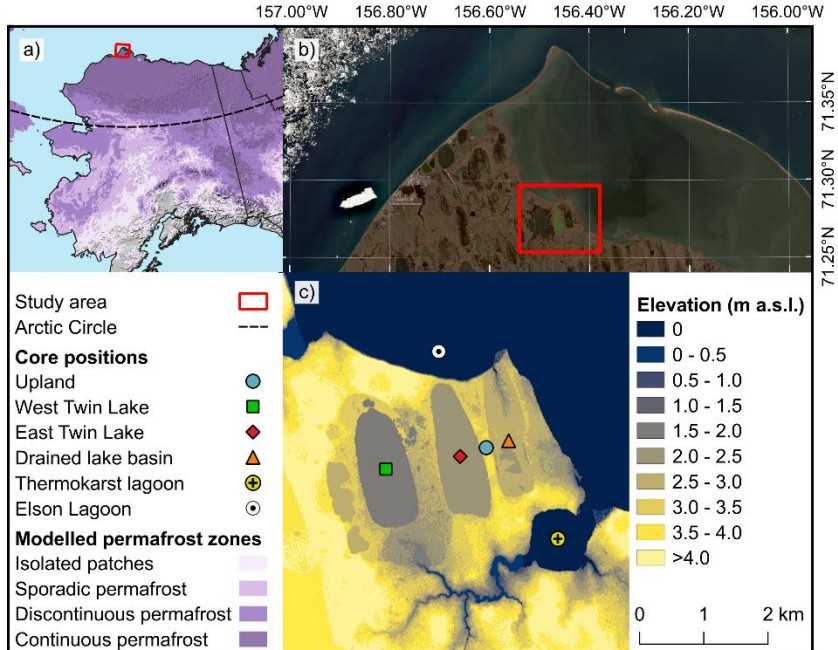

**Figure 1: The study area located close to Utqiaġvik at the North Slope of Alaska. (a) Alaskan permafrost extent map based on data from Obu et al. (2018), (b) Copernicus (2023) Sentinel-2 LA satellite image of the Barrow Peninsula and Elson Lagoon, and**
**(c) IFSAR digital terrain model (DGGS, 2018) with location of the coring sites. Map originally created by Lars Ebel.**

## 2.2 Methods

### 2.2.1 Fieldwork

Sediment coring took place in April 2022. As our objective was to decipher the transformation of organic carbon during permafrost thaw along a thaw and salt transect, coring positions were chosen form the least thaw and salt affected location,

the upland, to the most thaw and salt affected sites, the lagoons. Characteristics of core sites and metadata are provided in Table S1. Depending on the sediment conditions, three different coring devices were used. Frozen material from the upland and the DLB sites were cored with a "Snow, Ice and Permafrost Establishment" (SIPRE) corer. Since bedfast ice was present at Elson Lagoon and the semi-open thermokarst lagoon, sediments were cored with a SIPRE corer, too. For transportation the cores were wrapped in plastic foil and stored frozen in thermoboxes.

Unfrozen sediments of West Twin Lake were cored using a push corer (UWITEC gravity corer). Additionally, to get deeper, East Twin Lake was sampled with a vibration coring system (Livingstone-type drive rod piston corer). These cores were kept unfrozen in their PVC (push core) and aluminium (vibra core) tubes for transportation to AWI (Potsdam, Germany).



### 2.2.2 Laboratory work

**Sample preparation**


In preparation for the investigation of individual biogeochemical parameters, SIPRE cores were treated in a climate chamber at -8°C. The lake cores were prepared at 4 °C. All cores were cut in halves, cleaned, photographed, and described (sediment structures, color, cryostructures, Table S2-7). Subsequently, subsamples were divided in adjusted steps following the sediment structure and weighted.


**Depth correction for vibra core compaction**

During vibracoring of the East Twin Lake, especially sediments in the upper part of the lake deposits experienced compaction due to high water content and extended vibration time, while the lower sediments were more consolidated and

subjected to less compaction. To correct the recovered core depths for compaction, we applied a normalized exponential depth correction model, following Athy's law of exponential porosity–depth decay (Athy, 1930) and approaches used in ocean drilling studies (Lisiecki and Herbert, 2007). The true sediment depth $z_t$ was calculated from the measured depth $z_m$ (in the core liner) using the formula:

$$z_t = D_p \times \frac{1-e^{-z_m/\lambda}}{1-e^{-L_c/\lambda}} \tag{1}$$

where $D_p$ is the total penetration depth of the coring barrels (259 cm in this case), $L_c$ is the total recovered core length (here 138 cm), and $\lambda$ (lambda) is the exponential e-folding depth scale. The e-folding depth ($\lambda$) is defined as the characteristic depth over which a quantity decreases to 1/e (approximately 37%) of its initial value, assuming an exponential compaction. In this study, $\lambda$ represents the depth scale at which sediment compaction effectively reduces porosity or thickness by a factor of e due to vibration-induced consolidation. Based on previous studies of lakes, we adapt this with thermokarst properties

(especially water content, grain size, and porosity, e.g., Walter Anthony et al., 2014), and assumed $\lambda = 50$ cm, reflecting the highly compressible, water-rich upper layers. This correction ensures that the calculated true depth equals $D_p$ at the base of the core and follows an exponential compaction trend consistent with observed porosity–depth profiles in similar sediments.

**Porewater extraction and electrical conductivity measurements**


All samples were continued to be treated at 4 °C. Porewater was extracted with Rhizon samplers (0.12-0.18 µm membrane pore size). The extracted waters were measured with an Orion VERSASTAR PRO (Thermo Fisher) for electrical conductivity (mS cm⁻¹). We adapted salinity stages from Cahyadi et al. (2018) as follows: Freshwater < 1.5 mS cm⁻¹, brackish 1.5-15.0 mS cm⁻¹, saline 15.0-50.0 mS cm⁻¹, and hypersaline > 50 mS cm⁻¹. After porewater extraction, the





sediment remains were freeze-dried, and the water/ice contents (in weight %, wt%) were calculated by taking into account the wet and dry sample weights.

**Sedimentology and elemental analyses**

For the investigation of the grain size distribution, subsamples were treated with hydrogen peroxide for the removal of organic matter for four weeks. The grain sizes were measured with a Malvern Mastersizer 3000 (0.01–1000 µm grain size range) and evaluated with GRADISTAT 8.0 (Blott and Pye, 2001).

Using further subsamples, samples were milled and homogenized for the elemental analyses using a planetary mill (Fritsch PULVERISETTE 5). Using a soliTOC and a rapid N exceed element analyzer, total organic carbon (TOC) and total nitrogen
(TN) contents were quantified through the combustion and the analysis of the resulting gases. The detection limits of TOC and TN are 0.001 wt% and 0.05 wt%, respectively, however, we set the detection limit artificially to 0.1 wt% for both elements. TOC and TN contents below the detection limit (0.1 wt%) were replaced by half of the detection limit (0.05 wt%). This to prevent a bias and is according to Strauss et al. (2022) to make the point that low measurements are not the same as no measurements, avoiding zero inflation. For the TN samples below the detection limit, no C:N ratio calculation was done
to avoid artificial artifacts. The C:N ratio is a common indicator of organic matter degradation, with lower values reflecting higher decomposition (e.g., Fuchs et al., 2019; Mu et al., 2025). It also helps infer source material, distinguishing algal input (C:N < 10) from terrestrial plant matter (C:N > 20) (Meyers, 1994).

**Stable organic carbon isotope analysis**


Stable carbon isotope ratios are commonly applied as a proxy for organic matter origin and degradation in permafrost regions (e.g., Alewell et al., 2011; Strauss et al., 2015). We applied $\delta^{13}C$ analyses to gain a deeper understanding of the sediment characteristics. The samples were decarbonized with hydrochloric acid and quantification of $\delta^{13}C$ was conducted with a ThermoFisher Scientific Delta-V-Advantage gas mass spectrometer equipped with a FLASH elemental analyser EA
2000 and a CONFLO IV gas mixing system. Ultimately, $\delta^{13}C$ is expressed in ‰ vs the Vienna Pee Dee Belemnite (Coplen et al., 2006).

**Radiocarbon dating**

Radiocarbon dating was conducted for 21 selected samples at the MICADAS radiocarbon laboratory (Mollenhauer et al., 2021) at AWI, Bremerhaven (Germany). Preferably, plant remains were picked for dating, however, bulk sediment was analyzed for four samples due to the lack of macrofossils (Table S8). The radiocarbon ages were calibrated with Calib 8.20 using the IntCal 20 calibration curve (Reimer et al., 2020; Stuiver and Reimer, 1993).





**_n_-Alkane biomarkers**

The total lipid extract from freeze-dried, milled and homogenized sediment was eluted with a Dionex ASE 350 Accelerated Solvent Extractor using dichloromethane/methanol (DCM/MeOH, 99:1 v/v, heating phase 5 min, static phase 20 min at 75°C and 10 MPa). Next, the internal standard 5α-androstane was added to the extract. Asphaltenes were precipitated to avoid

complications during the next step, where the _n_-alkane including aliphatic fraction was separated via medium pressure liquid chromatography (Radke et al., 1980). This fraction was then measured using a Thermo Scientific ISQ 7000 Single Quadrupole Mass Spectrometer in combination with a Thermo Scientific Trace 1310 Gas Chromatograph (capillary column from BPX5, 2 mm × 50 m, 0.25 mm). With a total run time of 118 min, the MS transfer line temperature was set to 250 °C and the ion source temperature to 230 °C (ionisation energy 70 eV at 50 µA). Targeted screening of mid- and long chain _n_-

alkanes ($C_{23}$-$C_{33}$) was realized with the software Xcalibur (Thermo Fisher). The mass spectra (_m/z_ 50-600 Da, 2.5 scans s$^{-1}$) and the internal standard were used for compound specific identification and quantification.

**_n_-Alkane proxies**

The total _n_-alkane content (TAC) is the first proxy that is indicative of (paleo) environmental conditions and transformation effects (e.g., Brittingham et al., 2017; Thomas et al., 2021). We provide the TAC for the following chain lengths in µg/g Sediment (Sed):

$$TAC = \Sigma C_{23} - C_{33} \tag{2}$$

The average chain length (ACL), developed by Poynter and Eglinton (1990), is a proxy that is indicative of the organic

matter source. Algae and microorganisms typically have short-chain _n_-alkanes (< $C_{22}$), aquatic (submerged/floating, e.g., _Sphagnum_ moss) vegetation can be recognized by a dominance of mid-chain _n_-alkanes ($C_{23}$/$C_{25}$) while terrestrial vegetation is dominated by long-chain _n_-alkanes (> $C_{25}$) (Baas et al., 2000; Ficken et al., 2000; Killops and Killops, 2005; Otto and Simpson, 2005). Terrestrial vegetation can further be distinguished between shrubs and trees ($C_{27}$/$C_{29}$ predominance) and grasses and herbs ($C_{31}$/$C_{33}$ predominance) (Maffei, 1996; Schäfer et al., 2016). This chemotaxonomic "fingerprint", as

Eglinton et al. (1962) put it, can therefore be used as a tool for past environmental changes (e.g., Schwark et al., 2002; Zech et al., 2010). A limitation of _n_-alkane biomarkers concerns their blindness towards gymnosperms (e.g., Schäfer et al., 2016; Zech et al., 2021). The ACL is calculated following Brittingham et al. (2017) and others:

$$ACL_{23-33} = \frac{\Sigma \, i \times C_i}{\Sigma \, C_i} \tag{3}$$

Odd chain alkanes predominate over even chain numbers (Eglinton et al., 1962). The ratio of odd against even chain lengths,

called CPI, was first developed by Bray and Evans (1965) and further developed by Marzi et al. (1993). With advancing organic matter decay this ratio decreases (e.g., Andersson and Meyers, 2012; Schäfer et al., 2016). The CPI values of algae,



bacteria, and highly degraded substances such as oil approach one while modern vegetation samples have been reported with CPI values of up to 82 (Diefendorf et al., 2011; Killops and Killops, 2005; Tipple and Pagani, 2010). We calculated the CPI following Giest et al. (2025) and others:

$$CPI_{23-33} = \frac{\Sigma\ odd\ C_{23-31} + \Sigma\ odd\ C_{25-33}}{2 \times \Sigma\ even\ C_{24-32}} \tag{4}$$

To differentiate terrestrial vegetation between trees and shrubs (approaching 0), and grasses and herbs (approaching 1), we applied the endmember model of Schäfer et al. (2016):

$$n - alkane\ ratio\ = \frac{C_{31} + C_{33}}{C_{27} + C_{31} + C_{33}} \tag{5}$$

Ficken et al. (2000) developed a proxy ($P_{aq}$) which allows to more specifically quantify the share of aquatic macrophyte
input into the sedimentary record. Submerged/floating macrophytes reveal relatively high values ($P_{aq} > 0.4$) compared to emergent and terrestrial plants ($0.1 < P_{aq} < 0.4$; $P_{aq} < 0.1$; respectively). Since we investigated marine and lacustrine sediments, we applied $P_{aq}$ accordingly:

$$Paq\ = \frac{C_{23} + C_{25}}{C_{23} + C_{25} + C_{29} + C_{31}} \tag{6}$$

### 2.2.3 Statistical approaches

The combination of the biogeochemical and hydrochemical parameters was used to interpret units along the sediment cores (described top-down, labelled in Roman numerals) to enhance clarity and facilitate understanding.

In order to statistically quantify the characteristics of organic matter parameters (TOC, C:N, $\delta^{13}$C) in different sediment regimes, we created groups of thermal conditions (seasonally frozen, perennially frozen, unfrozen, refrozen, marine) and salinity stages (freshwater, brackish, saline, hypersaline). Due to violations of normality in multiple groups, as determined by
Shapiro-Wilk tests, we used the non-parametric Kruskal-Wallis test to assess differences in TOC, C:N, and $\delta^{13}$C among groups. For subsequent pairwise comparisons of the categories with the parameters, Mann-Whitney U tests were applied, coupled with the Benjamini-Hochberg (BH) p-value adjustment method. *n*-Alkane proxies were not included in the statistical tests due to limited sample counts in individual groups.

To quantify if any group statistically differs from another based on the combination of TOC, C:N, and $\delta^{13}$C, we performed a
Permutational Multivariate Analysis of Variance (PERMANOVA). We also created unique groups based on both their thermal and salinity condition and tested these with a PERMANOVA. Following a significant overall result, we conducted post-hoc pairwise PERMANOVA tests with p-value adjustment (BH) to identify which specific groups differed. All statistical analyses were carried out in RStudio and AI tools have been utilized for R coding.





## 3. Results

### 3.1 Biogeochemistry of the individual cores

#### 3.1.1 Permafrost upland

The upland core from an undisturbed tundra site not affected by prior abrupt thawing and refreezing since deposition is characterized by silt-dominated ice-rich permafrost in the first two meters of the soil column (on average 66 wt% ground ice; Fig. 2a). The electrical conductivity of the porewater indicates freshwater conditions (< 1 mS cm$^{-1}$) and a trend towards saline conditions (max. 17.3 mS cm$^{-1}$) below 140 cm depth (Unit VI). At 75 and 102 cm depth, radiocarbon dating reveals similar ages with 6.98 and 6.97 cal ka BP, respectively, while a lower sample at 186 cm depth has an age of 13.263 cal ka BP. This suggests that Holocene deposits with a thickness of at least 126 cm cover the underlying Lateglacial sediments. The TOC contents range between 35 wt% (Unit I) and 5.6 wt% (Unit VI). Between 70 and 100 cm depth, TOC contents are increased (Unit IV) compared to sediments above and below. TN contents relate to TOC contents and range between 0.4 and 1.7 wt% (on average 0.7 wt%). Overall, the C:N ratio is relatively homogenous along the core (on average 17.1). In the upper meter (Unit I-IV), the ratio is increased (19.7) compared to the lower meter (14.4). Carbon isotopic values range from an average of -26.8 ‰ in the upper 48 cm (Units I, II) to -27.7 ‰ between 53 and 198 cm depth. C:N and $\delta^{13}$C (r = 0.78, p < 0.001) correlate significantly.

TAC in the upland indicates a decreasing trend with depth (average 42.0 µg/g Sed; Fig. 3a). The maximum content is found at 10 cm depth with 113.3 µg/g Sed and the minimum at 186 cm depth with 5.1 µg/g Sed. Long-chain *n*-alkanes dominate (average ACL 26.9) and the degradation proxy CPI (average 15.1) reveals a distinct pattern. First, the CPI increases with depth from 10.6 to 21.5 (75 cm depth). Then, it drops to 10.0 (126 cm depth) and increases again to 15.6 (186 cm depth). The *n*-alkane ratio increases along the core from 0.1 (uppermost sample) and 0.4 (lowermost sample). P$_{aq}$ indicates aquatic influences with values between 0.4 and 0.7.

#### 3.1.2 West Twin Lake

The pushcore of the West Twin Lake talik is 23 cm long and therefore represents the surface sediments of the thermokarst lake (Fig. 2b). The sediment consists of fine silt (mean grain size 5 µm) with porewater of slightly brackish conditions (2 mS cm$^{-1}$). TOC contents are on average 16 wt%, and TN values range from below detection limit to 0.9 wt%. This results in C:N ratios of 18 (8 cm depth), 17 (16 cm depth), and 19 (21 cm depth). The $\delta^{13}$C ratio is on average -28.6 ‰.

One biomarker sample was analyzed at the West Twin Lake core at 8 cm depth (Fig. 3b). The total alkane content is 192.2 µg/g Sed and is therefore higher than in the upland. On the other hand, the ACL (25.5) and CPI (6.9) are lower compared to the upland. The *n*-alkane ratio and P$_{aq}$ are 0.2 and 0.8, respectively.



### 3.1.3 East Twin Lake

The sediments in the vibracore from the East Twin Lake are sandy to silty (Fig. 2c). Until 187 cm depth (Unit I, II) mean
grain sizes increase to 69 µm, and below that they average 16 µm. The electrical conductivity continuously increases from
brackish (14 mS cm⁻¹ at 8 cm depth) to saline (43 mS cm⁻¹ at 235 cm depth). Subsequently, the conductivity decreases but
remains in saline conditions. At 41 cm depth, radiocarbon dating revealed a Holocene age of 2,920 cal ka BP. At 215 and
252 cm depth, the sediment is of late Pleistocene age, 27,730 cal ka BP and 42,566 cal ka BP, respectively (Unit III). At the
lake bottom, the sediment has a TOC content of 16 wt% (Unit I). Subsequently, TOC decreases and remains < 3 wt% below
123 cm depth (Unit II, III). Only six out of 26 samples showed TN values above the detection limit. The TN content in these
samples is 0.1-0.9 wt%. The resulting average C:N ratio is 9.9. Carbon isotopic values increase from a minimum of -28.9 ‰
at 41 cm depth to -25.7 ‰ at 197 cm depth. Below that, $\delta^{13}C$ values remain relatively homogenous until the core bottom at
258 cm with an average of -25.8 ‰. C:N and $\delta^{13}C$ (r = -0.91, p = 0.01) reveal a very strong negative correlation.

The surface sediment sample of East Twin Lake has a TAC of 305.4 µg/g Sed, while at 75 and 121 cm depth, it decreased to
1.9 and 17.0 µg/g Sed, respectively (Fig. 3c). Long-chain *n*-alkanes predominate (average ACL 26.0). CPI (average 6.2), *n*-
alkane ratio (average 0.3), and $P_{aq}$ (0.7) are similar as in West Twin Lake.

### 3.1.4 Drained lake basin

The DLB core comprises frozen silty sand (Fig. 2d). The permafrost is less ice-rich compared to the upland (on average 49
wt% ground ice), and the electrical conductivity indicates predominantly brackish porewater conditions until 110 cm depth
(on average 4 mS cm⁻¹). The basal surface peat (21 cm depth, Unit I), an indication of the drainage event date, has an age of
0.7 cal ka BP. Between 74 and 108 cm depth, an age-depth inversion occurs with 10.5 and 9.9 cal ka BP, respectively. The
first late Pleistocene age is found at 140 cm depth with 34.1 cal ka BP. TOC values range between 2 (129 cm depth) and 42
wt% (undecomposed surface peat). Between 74 and 113 cm depth (Unit IV), a cryoturbated sediment structure is found,
which is characterized by organic-rich sediments (34 wt% TOC at 113 cm depth). The TN contents are in the range of
contents found in the upland, but are on average slightly lower (0.5 wt%). The C:N ratio reflects the TOC and TN patterns
along the core however, peak values (0-15 cm and 113 cm depth) are less dominant. The $\delta^{13}C$ values increase with depth and
range between -29.4 and -25.0 ‰ at 2.5 and 136.5 cm depth, respectively. No significant correlation is found between C:N
and $\delta^{13}C$ (r = -0.35, p = 0.08).

Compared to all investigated landforms, *n*-alkane specific parameters indicate the strongest heterogeneity in the DLB (Fig.
3d). TAC ranges between 0.9 and 519.8 µg/g Sed, which is the lowest and highest observed value in our study, respectively.
The minimum ACL (25.6 at 31 cm depth) is similar to the surface samples of the twin lakes. At 108 cm depth, extreme
values are reached for ACL (30.2), CPI (51.3), the *n*-alkane ratio (0.9), and $P_{aq}$ (0.02).





**Figure 2: Sedimentological, elemental, isotopic and hydrochemical data from the investigated sediment cores. (a) UL (upland) , (b) WTL (West Twin Lake), (c) ETL (East Twin Lake), (d) DLB (Drained lake basin), (e) SOL (Semi-open lagoon), (f) EL (Elson Lagoon).**





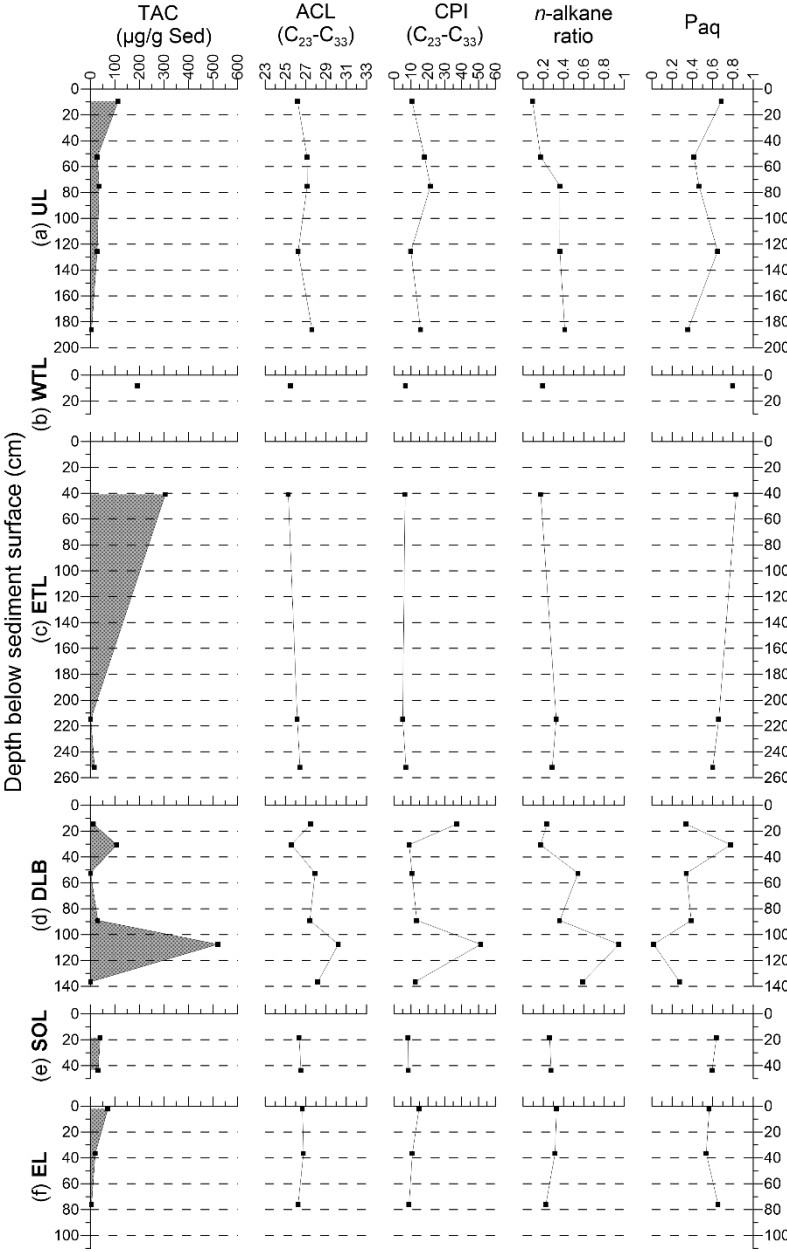

Figure 3: n-Alkane proxies along the cores before (gray) and after (red) the incubation. (a) UL (upland) , (b) WTL (West Twin Lake), (c) ETL (East Twin Lake), (d) DLB (Drained lake basin), (e) SOL (Semi-open lagoon), (f) EL (Elson Lagoon).

**3.1.5 Semi-open thermokarst lagoon**

The thermokarst lagoon core contains silty sediments (Fig. 2e). The average water content of 44 wt% and the porewater is characterized by hypersaline conditions with a peak value 119 mS cm⁻¹ at 34 cm depth. Radiocarbon dating of plant macrofossils shows an age-depth inversion with 7.2 cal ka BP at 19 cm depth and 4.6 cal ka BP at 44 cm depth. The TOC



and TN contents average 6 wt% and 0.4 wt%, respectively. This results in an average C:N ratio of 16. Relatively homogenous values are also found for $\delta^{13}C$ with an average of -27.6 ‰. C:N and $\delta^{13}C$ (r = -0.50, p = 0.17) are not correlating significantly.

The two analyzed biomarker samples in the semi-open lagoon are similar (Fig. 3e). A TAC of 40.0 and 31.5 µg/g Sed is present at 19 and 44 cm depth, respectively. The average ACL (26.4) and CPI (8.3) lie between the observed values of the thermokarst lakes and the terrestrial sites. Also, the average *n*-alkane ratio (0.3) and $P_{aq}$ (0.6) are within the range of values observed in the other cores.

### 3.1.6 Elson Lagoon

The Elson Lagoon core comprises silty sediments underlain by sand (Fig. 2f). The maximum and minimum porewater contents are 56 and 17 wt% at 7 and 109 cm depth, respectively. Compared to the thermokarst lagoon, Elson Lagoon is less saline, yet hypersaline conditions prevail (on average 76 mS cm$^{-1}$). Radiocarbon ages signalize sediment mixing as plant macrofossils have an age of 5.5 cal ka BP at 2 cm depth and 5.3 cal ka BP at 37 cm depth. Along the core, TOC contents are on average 4 wt% and show a decreasing trend. Until 87 cm depth, TN averages 0.3 wt%, and contents are < 0.1 wt% below this depth. The C:N ratio is relatively homogenous with an average of 18. $\delta^{13}C$ lies on average at -27.7 ‰ until 76 cm depth (Unit I). From 81 to 109 cm depth (Unit II) $\delta^{13}C$ precipitously increases to -25.4 ‰, which is the maximum value compared to all cores. No significant correlation is found between C:N and $\delta^{13}C$ (r = -0.26, p = 0.32).

The sediments of Elson Lagoon reveal a decreasing trend of TAC with depth (average 32.1 µg/g Sed; Fig. 3f). *n*-Alkane proxies compare generally well to the thermokarst lagoon with the exception of the CPI. The degradation proxy indicates less degraded material (average 11.4) in the non-thermokarst lagoon compared to the thermokarst lagoon. Otherwise, average ACL (26.6), *n*-alkane ratio (0.3), and $P_{aq}$ (0.6) are similar as in the semi-open lagoon.

## 3.2 Influence of temperature and salinity

### 3.2.1 Thermal sediment stages

Our five thermal sediment stages comprise *seasonally frozen* (max. active layer depths of UL and DLB after Nyland et al. (2021)), perennially *frozen* (permafrost domain of UL), *refrozen* (DLB sediments), *unfrozen* (talik/cryopeg of West and East Twin Lake), and *lagoon* (lagoon sediments) (Table 1).

Kruskal-Wallis tests show that significant differences exist among the thermal regimes for TOC (p < 0.001), C:N (p < 0.001), and $\delta^{13}C$ (p = 0.02). Significance levels of pairwise Mann-Whitney test results are visualized in Fig. 4. The PERMANOVA test considering the combination of TOC, C:N and $\delta^{13}C$ indicated significant differences between thermal regimes (r = 0.40, p = 0.001). Post-hoc testing revealed that *unfrozen* sediments are significantly different from *frozen* (r = 0.40, p = 0.05), *refrozen* (r = 0.55, p = 0.02), and *lagoon* sediments (r = 0.40, p = 0.04). *Frozen* sediments further differ from




*refrozen* (r = 0.40, p = 0.03) and *lagoon* (r = 0.55, p = 0.01) sediments. Also, significant differences exist between *refrozen*

and *lagoon* sediments (r= 0.40, p = 0.03).

**Table 1: Grouping of sediment samples into thermal stages. Sample count (*n*) in individual groups with reduced number after n/a removal in brackets. UL - Upland, WTL - West Twin Lake, ETL - East Twin Lake, DLB - Drained lake basin, SOL - Semi-open lagoon, EL - Elson Lagoon.**

| Thermal stage | UL | | WTL | | ETL | | DLB | | SOL | | EL | | total |
|---|---|---|---|---|---|---|---|---|---|---|---|---|---|
| | *n* | depth (cm) | *n* | depth (cm) | *n* | depth (cm) | *n* | depth (cm) | *n* | depth (cm) | *n* | depth (cm) | *n* |
| seasonally frozen | 8 | 3-43 | | | | | 9 | 0-47 | | | | | 17 |
| perennially frozen | 26 | 48-198 | | | | | | | | | | | 26 |
| unfrozen | | | 4 (3) | 2-21 | 26 (6) | 8-258 | | | | | | | 30 (9) |
| refrozen | | | | | | | 17 | 53-118 | | | | | 17 |
| lagoon | | | | | | | | | 10 (9) | 3-131 | 21 (17) | 2-109 | 31 (26) |

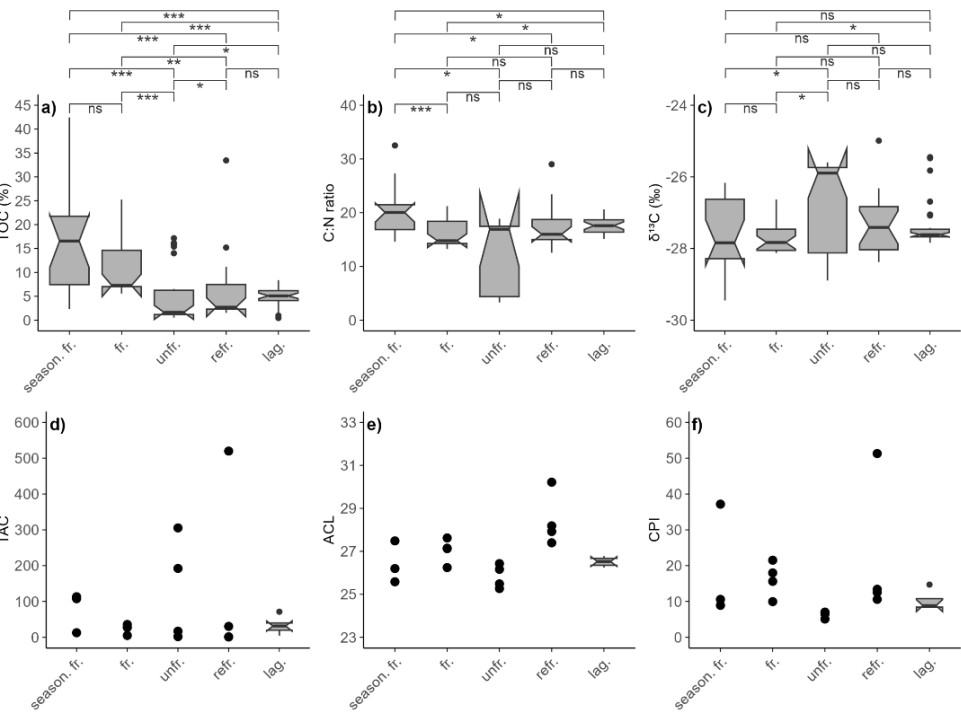

**Figure 4: Notched boxplots of organic carbon parameters based on thermal regimes. Boxplots are shown for sample counts >4. Sediment classes comprise seasonally frozen (season. fr.), frozen (fr.), unfrozen (unfr.), refrozen (refr.) and lagoon (lag.).**



**Significance brackets are plotted for (a) TOC, (b) C:N (c) δ13C, and comprise the significance levels p < 0.001 (\*\*\*), p < 0.01 (\*\*), p < 0.05 (\*) and p > 0.05 (ns).**

### 3.2.2 Salinity stages

Grouping of sediment samples was conducted after previously defined salinity stages (section 2.2.2, Table 2).

Significant differences in TOC (p < 0.001), C:N (p < 0.001), and δ¹³C (p < 0.001) exist across at least some of the defined

salinity stages according to Kruskal-Wallis. Pairwise Mann-Whitney test significant levels are plotted in Fig. 5.

PERMANOVA testing indicated that significant differences in salinity categories exist (r = 0.55, p = 0.001). Post-hoc testing

revealed weak but significant differences between *brackish* and *hypersaline* samples (r = 0.07, p = 0.05).

**Table 2: Grouping of sediment samples into salinity stages: Freshwater < 1.5 mS cm⁻¹, brackish 1.5-15.0 mS cm⁻¹, saline 15.0-50.0**
**mS cm⁻¹, and hypersaline > 50 mS cm⁻¹. Sample count (*n*) in individual groups with reduced number after n/a removal in brackets.**
**UL - Upland, WTL - West Twin Lake, ETL - East Twin Lake, DLB - Drained lake basin, SOL - Semi-open lagoon, EL - Elson**
**Lagoon.**

| Salinity stage | UL (*n*) | WTL (*n*) | ETL (*n*) | DLB (*n*) | SOL (*n*) | EL (*n*) | total (*n*) |
|---|---|---|---|---|---|---|---|
| freshwater | 24 | | | 3 | | | 27 |
| brackish | 9 | 4 (3) | 2 | 19 | | | 34 (33) |
| saline | 1 | | 24 (4) | 4 | | | 29 (9) |
| hypersaline | | | | | 10 (9) | 21 (17) | 31 (26) |

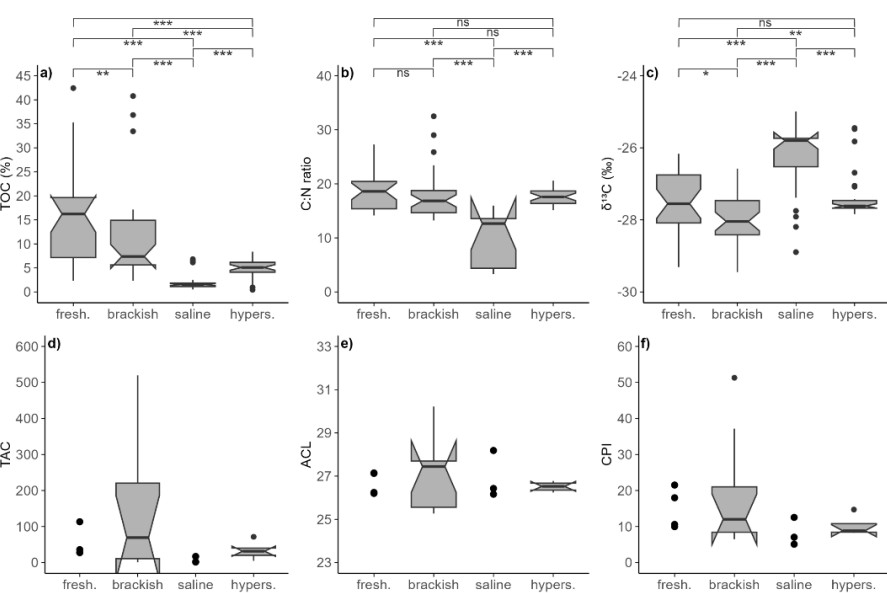

**Figure 5: Notched boxplots of organic carbon parameters based on salinity categories. Boxplots are shown for sample counts >4.**
**Significance brackets are plotted for (a) TOC, (b) C:N, (c) δ13C and comprise the significance levels p < 0.001 (\*\*\*), p < 0.01 (\*\*),**
**p < 0.05 (\*), and p > 0.05 (ns).**



### 3.2.3 Combined effect of thermal and salinity regimes

Combining the thermal and salinity stages into unique groups resulted in eleven groups (Table 3). In PERMANOVA tests specific groups were excluded from the analysis due to low sample counts (n < 5 in G5, G7, G8, G10, Table 3). The PERMANOVA indicated that groups differ significantly from each other concerning TOC, C:N and $\delta^{13}$C (r = 0.58, p = 0.001). Post-hoc pairwise PERMANOVA tests revealed that Group 2 and Group 6 were significantly different (r = 0.87, p = 0.03). Also, Group 3 differed significantly from Group 9 (r = 0.87, p < 0.01).

**Table 3: Grouped thermal and salinity regimes and contributing cores. Sample count (*n*) in individual groups with reduced number after n/a removal in brackets. Asterisks mark small group sizes (< 5). These groups were not considered in the PERMANOVA analyses. UL - Upland, WTL - West Twin Lake, ETL - East Twin Lake, DLB - Drained lake basin, SOL - Semi-open lagoon, EL - Elson Lagoon.**

| Thermal regime | Salinity stage | Group | Contributing core | *n* |
|---|---|---|---|---|
| seasonally frozen | freshwater | G1 | UL, DLB | 9 |
| seasonally frozen | brackish | G2 | DLB | 8 |
| frozen | freshwater | G3 | UL | 16 |
| frozen | brackish | G4 | UL | 9 |
| frozen | saline | G5 | UL | 1* |
| unfrozen | brackish | G6 | WTL, ETL | 6 (5) |
| unfrozen | saline | G7 | ETL | 24 (4)* |
| refrozen | freshwater | G8 | DLB | 2* |
| refrozen | brackish | G9 | DLB | 11 |
| refrozen | saline | G10 | DLB | 4* |
| lagoon | hypersaline | G11 | SOL, EL | 31 (26) |

## 4. Discussion

### 4.1 Modern and paleoenvironmental processes and conditions

The biogeochemistry of the investigated landforms provides insights into the landscape history and current environmental conditions. The recent active layer depth in uplands ranges between 29 cm and 47 cm (Nyland et al., 2021). This aligns well with the increased TOC and TN contents of Unit I (Fig. 2a). Our measured electrical conductivity in the sediments reflects freshwater conditions which were also quantified by Jones et al. (2023) in the same permafrost upland and that are found in nearby streams (Lougheed et al., 2020). Unit II represents the range of the late-season thaw depth and the transient layer,



which can be recognized by increased water/ice contents (Shur et al., 2005). The permafrost begins below 47 cm (Unit III).
       Bockheim et al., (1999) found that > 75 % of soils in the region are affected by cryoturbation, and signs for this are found in
       the upland core between 70 and 100 cm depths (Unit IV, Table S2). TOC and TN contents are increased compared to the
       units above and below, and the radiocarbon dates reveal an age-depth inversion. Cryoturbation at this depth could point
       towards a paleo active layer from the mid Holocene. Relatively homogenous conditions in the second meter (Unit V, VI)
indicate fairly stable conditions at the upland site from the Lateglacial into the early Holocene period.

       Through the investigation of pollen records, Meyer et al. (2010) describe a shift from a purely grass dominated tundra in the
       Lateglacial towards a grass, sedge, and dwarf shrub mix in the early Holocene. Considering the ACL (average 26.9) and *n*-
       alkane ratio (< 0.4), it could be inferred that this slowly occurring shrubification throughout the Holocene can be confirmed
       by our *n*-alkane analysis (Fig. 3a). Considering $P_{aq}$ as well, this indicates that aquatic vegetation increasingly influences the
ACL. Aquatic vegetation (e.g., *Sphagnum* sp.) is present in ice-wedge troughs (Lara et al., 2015), yet $P_{aq}$ has to be
       interpreted with caution. *Betula* shrubs, for example, are reported with a mid-chain *n*-alkane predominance (Weber and
       Schwark, 2020) which would impact $P_{aq}$. Nevertheless, the modern and paleo vegetation composition is likely reflected by
       our *n*-alkane record.

Different from the gradual top-down thaw processes in the upland, thermokarst lakes are affected by abrupt thaw processes
       which can reach much deeper deposits over short time scales (Grosse et al., 2011; Webb et al., 2025). While uplands and
       thermokarst lakes are sedimentologically connected due to the same parent material, the biogeochemical fingerprints of lakes
       may have changed due to recent thaw processes and lacustrine deposition. Comparing our two lake sites is challenging, as
       the West Twin Lake sediment core is very short, thus, a full comparison to the longer East Twin Lake core is somewhat
limited. The main difference is in the salinity between the lake sediments. While brackish conditions occur beneath West
       Twin Lake, the sediments of East Twin Lake have predominantly saline porewater (Fig. 2b, c; Table 2). As negative
       temperatures were measured in the sediments of East Twin Lake (Jones et al., 2023), we define the unfrozen sediments as a
       cryopeg. As Jones et al. (2023) reports, West Twin Lake retains freshwater conditions, yet we found brackish talik
       sediments. A near-future shift towards brackish lake water is therefore very likely, as East Twin Lake has already
experienced. Along the East Twin Lake core, lake sediments (Unit I) can be distinguished from the underlying late
       Pleistocene marine sediments (Unit III) (Eisner et al., 2005). The lake sediments cover ca. 123 cm as inferred from increased
       TOC contents. These are in the range of values observed in the Teshepuk Lake area (Lenz et al., 2016). $P_{aq}$ indicates an
       increased aquatic influence in the upper sediments. Yet ACL and the *n*-alkane ratio also indicate terrestrial plant input into
       the lake which can be explained by enhanced lateral expansion (shoreline erosion) in recent years (Jones et al., 2023). At this
point, it needs to be stressed that in our study the ACL decreases with decreasing CPI values (r = 0.79, p < 0.001), meaning
       that the vegetation signal is influenced by organic matter degradation (strongest in ETL). This is a commonly observed
       process (e.g., Jongejans et al., 2020; Struck et al., 2018), which needs consideration when interpreting *n*-alkane proxies.



The transition from Unit II to III is characterized by a sharp decrease of the mean grain size from fine sand to silt. The sand might be aeolian which deposited in the surface depression (Carter, 1981; Eisner et al., 2005). In both units, the C:N and

$\delta^{13}$C ratios point towards a marine/lacustrine algae production (Fig. 4b, c; Meyers, 1994), which aligns with the high salinity and the radiocarbon dates (Figure 2c, Table S8). This pattern matches with the investigation of Meyer et al. (2010), who observed more enriched $\delta^{13}$C values in the salty late Pleistocene sediments at a permafrost tunnel on the Barrow Peninsula.

In the view of landscape alteration, the next step is the drainage of thermokarst lakes with permafrost aggregation at

subaerial conditions. The DLB sediment core revealed mixed signals as it covers characteristics from both the upland and the thermokarst lakes. On the one hand, the electrical conductivity is similar to the upland, which is linked to Holocene ages. On the other hand, the sediments are sandier as is the case for the East Twin Lake sediments. Brown et al. (2003) estimated that the lake drained about 100-150 years ago, however, our radiocarbon dating revealed that post-drainage peat formation was initiated ca. 700 cal a BP. DLBs sequester organic carbon in the form of peat (Bockheim et al., 2004; Hinkel et al., 2003;

Jones et al., 2012), which can be seen in our core with the highest TOC contents and C:N ratios in the upper 15 cm of all investigated cores (Unit I). We argue that the most recent lake phase in the DLB started 1.3 cal ka BP and therefore lasted only approximately 600 years when considering the TOC contents (Unit II). Between 21 and 31 cm depth, TOC contents are in the range of contents found in the East Twin Lake sediments, and $P_{aq}$ signalizes aquatic conditions. Our observation of the former lake phase shows that these periods are in this case shorter than previously estimated by Fuchs et al. (2019), who

found that lakes persisted a minimum of 1000 years on the coastal plain in northern Alaska. The relatively carbon-poor and alkane-depleted sediments beneath the lake phase might be an *in-situ* refrozen talik starting beneath the modern active layer (Unit III). Increased TOC contents and a high *n*-alkane ratio in Unit IV indicate one or even two wetland periods with organic matter accumulation, which likely occurred during the Holocene Thermal Maximum (Jones and Yu, 2010; Kaufman et al., 2004). The low TOC and TN content at 93 cm depth make it difficult to clarify whether a first wetland period was

followed by another, as this impression could also result from cryoturbation. The age-depth inversion is an indication of both a refrozen talik and intensified freeze-thaw dynamics during the early Holocene. Unit V represents the late Pleistocene to early Holocene transition as indicated by the trend towards lighter $\delta^{13}$C values (Fig. 2d). Brackish refrozen sediment could thereby be statistically distinguished from frozen upland deposits affected by freshwater conditions (Section 3.2.3).

If drained basins or lakes are connected and inundated by sea water, they become thermokarst lagoons. With a size of 80 ha, the semi-open lagoon is a relatively small thermokarst lagoon based on the circum-Arctic thermokarst lagoon assessment published by Jenrich et al. (2025b). Both the TOC and TN contents are higher compared to other lagoons, such as on the Bykovsky Peninsula in Siberia (Jenrich et al., 2021; Schirrmeister et al., 2018; Ulyantsev et al., 2017; Yang et al., 2023). Shoreline erosion and fluvial organic matter input from streams (Fig. 1) are likely driving factors for these comparably high

TOC and TN contents.





Elson Lagoon, which is not a thermokarst lagoon, receives terrestrial sediment input through coastal erosion, and deposition takes place as the barrier islands protect the waters of the lagoon (Brown et al., 2003; Ping et al., 2011; Zimmermann et al., 2022). Our investigations indicate that the upper 80 cm of the sediments are terrestrial deposits, since the biogeochemical parameters are overall very similar to the values found in the sediments of the thermokarst lagoon (Unit I; Fig. 2f, 3f). These

are underlain by likely late Pleistocene to early Holocene sandy sediments (Unit II).

The age-depth inversions signalize sediment mixing in both lagoons, which is favored through the semi- to unfrozen conditions (Table S1). The hypersalinity in the sediments might be the result from concentrating salt through bedfasting ice in the shallow lagoons (Jenrich et al., 2021). Hypersaline conditions in Elson Lagoon were quantified earlier by Osterkamp and Harrison (1985) and Overduin et al. (2012).

**4.2 Organic matter degradation patterns**

For the terrestrial sampling measurements in the upland, the C:N ratio compares well to values reported in similar sites in northern Alaska (Fuchs et al., 2019; Giest et al., 2025; Ping et al., 2011). The comparably lower ratio in the second meter (Unit V, VI) points towards a slightly higher degradation state (Fig. 2a). C:N ratios are lower compared to permafrost-affected peat (Andersson et al., 2012) but higher than in Yedoma deposits (Strauss et al., 2022). Since $\delta^{13}C$ values become

lighter (less negative) with degradation, C:N should correlate negatively with $\delta^{13}C$ (Strauss et al., 2015). In our upland core however, this relationship shows a significant positive correlation. Since the CPI shows mixed but high (> 10) values, we can conclude that organic matter is overall relatively well preserved in the upland. PERMANOVA test results indicated that *frozen* sediments are distinct from sediments affected by thaw processes, namely *unfrozen*, *refrozen* and *lagoon* sediments. This expresses the possibility of statistically distinguishing between sediments of various thaw histories.

For the first degradation stages with the subaquatic lake phase, PERMANOVA testing also revealed a distinct biochemical signal of *unfrozen* sediments when compared to *frozen*, *refrozen*, and *lagoon* sediments. This result, as for the upland, is likely influenced by source signals, especially when terrestrial and marine deposits are compared. But since the CPI in the lakes (on average 6.4) is comparably lower than in the upland (on average 15.1) and the DLB (on average 22.3), the organic matter generally seems to be more degraded in the thermokarst lakes (Fig. 3). Moreover, the CPI in the saline East Twin

Lake expressed a stronger degradation signal (6.5) than the brackish West Twin Lake sediments (6.9). In comparison to Yedoma and other thermokarst lake sediments, the CPI indicates similar (Jongejans et al., 2020) to stronger (Jongejans et al., 2021) organic matter decomposition.

For the DLB, the organic carbon quality shows fresh organic matter in the surface peat (Unit I) which, together with the surface sediments of the upland, makes *seasonally frozen* organic-rich sediments distinguishable from all other thermal

regimes considering C:N as shown by the Mann-Whitney test (Fig. 4b). This is followed by more degraded conditions in the refrozen talik sediments (Unit II, III) and well-preserved organic carbon in the former wetland phase (Unit IV). The C:N ratio and the CPI index agree in their patterns, however, $\delta^{13}C$ only poorly reflects this picture (Fig. 2d, 3d). Therefore, C:N and $\delta^{13}C$ resulted in a weak and near-significant correlation (r = -0.35, p = 0.08). Especially the peak $\delta^{13}C$ values (least



negative) in Unit I and IV disturb the relationship, and it is likely that the source (i.e., lacustrine) signal of the isotopes is

responsible for this pattern. Unit V indicates comparably stronger degraded organic matter with an average C:N of 13.7 and a CPI of 12.5, combined with a marine $\delta^{13}$C signal (maximum -25.0 ‰). The unique pattern of refrozen sediments is reflected in the PERMANOVA results as the *refrozen* sediments could be differentiated from *frozen* and *unfrozen* terrestrial sediments as well as *lagoon* deposits.

Our sediment surface C:N and $\delta^{13}$C values fit into the range observed by Wolter et al. (2024). Comparing the average C:N of

the full core (18.6) to other study areas in northern Alaska and northwest Canada however indicates less degraded conditions than observed by Giest et al. (2025) (17.5), Fuchs et al. (2019) (16.6) and Wolter et al. (2017) (13.2). The increased C:N ratios from Unit IV are pivotal reasons for this result, as early wetland phases were not described in the aforementioned studies.

For the lagoons, the organic carbon degradation proxies reflect relatively little variability within the sediments and represent

a mixed signal from the terrestrial and lacustrine sites (Fig. 2, 3, 4). Although individual statistical tests of hypersaline lagoon sediments revealed differences to other sediment types (section 3.2.1 and 3.2.2), the combined effect of thermal and salinity regimes in lagoon sediments did not differ significantly from other sediment types (section 3.2.3). The reasons for this are the various sediment types that are mixed in the investigated lagoons. The C:N ratio and CPI indicate relatively stronger degraded material in the semi-open lagoon compared to Elson Lagoon, though compared to lagoons in northern

Alaska and Siberia, both exhibit well-preserved organic carbon, which can be explained by their terrestrial origin (Giest et al., 2025; Jenrich et al., 2021; Schirrmeister et al., 2018; Yang et al., 2023).

### 4.3 Implications and outlook

Our biogeochemical investigations into representative landforms of northernmost coastal Alaska provide a valuable basis for evaluating potential implications and future developments of the region. Based on this, we drew a conceptual model of

landscape dynamics in the study area (Fig. 6).

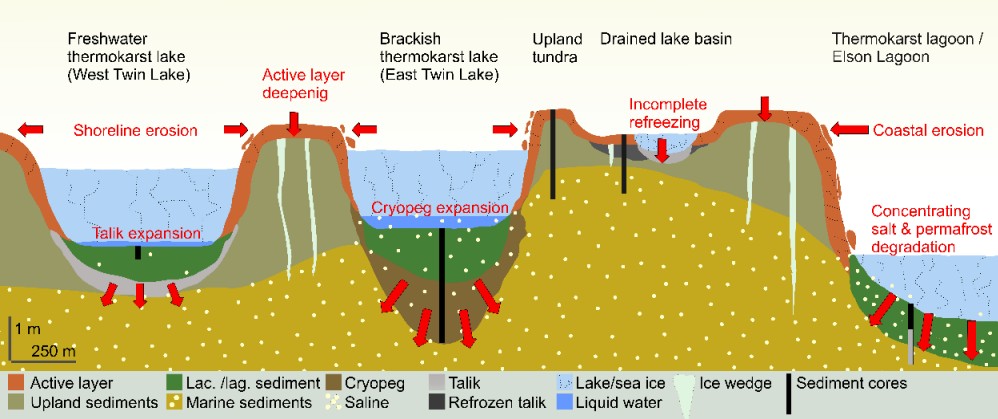

**Figure 6: Conceptual illustration of physical properties and dynamics currently affecting the Barrow Peninsula with potential for future acceleration. The sediments are present in a variety of thermal states, seasonally frozen (active layer), perennially frozen**



(upland and marine sediments), unfrozen (talik, lacustrine sediments, partly DLB), unfrozen cryotic (cryopeg and
lacustrine/lagoon sediments) and refrozen (DLB). Model illustrated at approximate scale.

Remnant uplands cover ca. 28 % of today's landscape in the study area (Hinkel et al., 2005; Jones et al., 2022) and saline permafrost underlays Holocene deposits (Brigham-Grette and Hopkins, 1995; Brouchkov, 2003; Eisner et al., 2005). These non-saline sediments have a thickness of less than 1.5 m in our investigated upland. Increasing active layer depths in undisturbed tundra (Liu et al., 2024; Smith et al., 2022) cause the mobilization and (dominantly) aerobic mineralization of
organic carbon, leading to $CO_2$ production (Jenrich et al., 2024). Subaerial talik formation is projected to propagate into the continuous permafrost zone within the 22nd century (Parazoo et al., 2018), exposing deep deposits to organic carbon alteration. Once thaw reaches to depths of the saline deposits, degradation processes will be enhanced as the thaw point is depressed. This poses the risk that subaerial taliks and cryopegs will be initiated earlier than expected.

East Twin Lake has already undergone this development turning it from a freshwater to a brackish/saline thermokarst lake,
which led to increased thermoerosion rates (Jones et al., 2023). Brown et al. (2003) calculated that the lake will likely drain into Elson Lagoon (or turn into a thermokarst lagoon) by the 2040s. However, this estimate was purely based on lake and lagoonal shoreline erosion rates while other processes, such as sea level rise and widespread gradual thaw with continued warming (Creel et al., 2024; Guimond et al., 2021) were not accounted for. These processes further increase the forcing, potentially leading to faster-than-anticipated landscape changes. The timing of a potential future drainage event of West
Twin Lake is more difficult to estimate, but the drainage pathway may be via the thermoerosional gully to the south of the lake (Fig. 1). A supra-permafrost groundwater connection between the lake and Elson Lagoon may also play a (future) role in groundwater discharge (Dimova et al., 2015; Rawlins, 2021), yet Guimond et al. (2022) conclude that groundwater is not a significant factor at the northern coast of Alaska due to a low land to sea hydraulic gradient. The talik beneath West Twin Lake has already reached brackish deposits, and with continued thaw beneath the lake, it can be expected that the lake's
water will turn brackish in the near future, too. This in turn would have implications for lake ice dynamics and continued thaw beneath the lake. Arp et al. (2012) observed a trend from bedfast towards floating lake ice conditions in northernmost Alaska, but the impact of saline conditions was not accounted for. Considering the vulnerability of permafrost to salt, it can be expected that the aforementioned cascading effects of salt intrusion into lake waters will play a crucial role across the Younger Outer Coastal Plain. The organic deposits within the lake's sediments are already relatively stronger degraded than
in the upland, and the sediments of the cryopeg in East Twin Lake are more heavily degraded than in the West Twin Lake talik. From a microbiological point of view, this can be explained by microbial communities that are likely dominated by methanogens ($CH_4$ producers) in West Twin Lake while in East Twin Lake sulfate-reducing microbes ($CO_2$ producers) probably established additionally due to the saline deposits, leading to enhanced mineralization (Jenrich et al., 2024; Yang et al., 2023). This comparison stresses the potential for year-round saline carbon degradation at sub-zero temperatures.

Northernmost Alaska is a highly dynamic region which is experiencing a net thermokarst lake area loss trend mostly due to drainage of large lakes (Nitze et al., 2017; Webb et al., 2022). A detailed analysis of pond dynamics within DLBs on the Barrow Peninsula could also verify a drainage trend (Andresen and Lougheed, 2015). We investigated a wet but terrestrial



site within the DLB with a refrozen talik. Still, two larger thermokarst ponds are situated nearby with potential taliks beneath. Thus, although the basin sequesters peat, it can be expected that carbon degradation also takes place locally.

Especially the former wetland phase (Unit IV, Fig. 2d, 3d) indicates well preserved organic matter in the sediment, holding a high potential for carbon degradation. If these ponds drain, it can be expected that permafrost aggradation will progressively slow with continued Arctic warming (Jones et al., 2022). Moreover, Wolter et al. (2024) quantified high $CH_4$ concentrations in submerged areas within DLBs showcasing the potential for carbon degradation under current environmental and climatic conditions.

When terrestrial or lacustrine deposits transition into, or are eroded into, lagoon systems (as in our cases), thawed organic carbon becomes vulnerable to decomposition under now hypersaline conditions. We found that the lagoonal deposits represent a mixed signal of the terrestrial and lacustrine sediments rather than further advanced carbon degradation. With increasing marine influence, it can be expected that greenhouse gas production shifts from mixed $CH_4$ and $CO_2$ in low connected water bodies (such as the semi-open lagoon) towards pure $CO_2$ production in more marine settings (like Elson

Lagoon, Jenrich et al., 2025a). Submarine permafrost degradation rates of up to 4 cm per year in Elson Lagoon (Overduin et al., 2012) also contribute to enlarging the carbon pool available for mineralization. The high content of TOC and its relatively well-preserved state therefore exhibits a large potential for future carbon mineralization in lagoon systems.

**5. Conclusions**

Our study reveals distinct patterns of organic carbon alteration along gradients of thaw and salinity stages in the Arctic

coastal lowland.

Regarding research question 1, which addressed how paleoenvironmental and modern processes shape present-day sediment characteristics on the Barrow Peninsula, we found that Lateglacial saline deposits are overlain by organic-rich, well-preserved sediments from an early Holocene wetland phase and a mid-Holocene active layer, as observed in the DLB and the remnant upland, respectively. Thermokarst lake subsidence has transformed these landscapes, with West Twin Lake

reaching brackish conditions and East Twin Lake progressing towards brackish/saline states. Biogeochemical signatures indicate ongoing input of terrestrial and lacustrine sediments into the lagoons.

In response to research question 2, which focused on organic matter quality in thermokarst landforms, we conclude that organic carbon degradation is enhanced in the brackish talik of West Twin Lake and particularly in the cryopeg of East Twin Lake. Lagoonal sediments show a high potential for carbon mineralization, driven by salt accumulation beneath bedfast ice

that has created hypersaline conditions. These conditions have maintained year-round unfrozen zones, enabling microbial activity and greenhouse gas production.

Concerning research question 3, which asked how landscape dynamics affect organic matter mobilization, we anticipate that deepening active layers, the formation of taliks and cryopegs, and ongoing shoreline erosion in lakes and lagoons will increase the risk of organic carbon mobilization and mineralization. This risk is especially pronounced in saline deposits and

may accelerate landscape change across the saline permafrost region, with substantial implications for permafrost carbon release.

Our findings show a need for additional local investigations and upscaling efforts assessing the consequences of saline permafrost thaw. While focused on a specific site on the northernmost tip of the USA, our study captures processes and conditions that are characteristic of extensive saline permafrost landscapes across the Arctic, underscoring its significance

for regional-scale assessments of carbon dynamics.

**Data availability**

The data used for this study is available in the PANGAEA open access archive under:

https://doi.pangaea.de/10.1594/PANGAEA.983965 (biogeochemical and hydrochemical data) and https://doi.pangaea.de/10.1594/PANGAEA.983966 (*n*-alkane data).

**Author Contributions**

FS and JS designed the study. JS, GG, MJ, and BMJ conducted fieldwork. FS, MJ, and JS carried out sediment subsampling. FS conducted laboratory work, analyzed and plotted data, and wrote the first manuscript. All authors contributed to the development of the paper.

**Competing interests**


Some authors are members of the editorial board of EGU Biogeosciences.

**Acknowledgements**

The SIPRE corer for this work was gratefully provided by Kenneth M. Hinkel (University of Cincinnati) and the vibracorer was gratefully provided by Chris Mayo (UAF). We also thank the Ukpeaġvik Iñupiat Corporation (UIC) for issuing the permit for this research and providing logistical support. We would like to thank the AWI laboratories Permafrost Biogeochemistry, Permafrost Hydrochemistry, ISOLAB and MICADAS for their support. This work is a contribution to the IPA Saline Permafrost Action Group.


**Financial support**





FS and MJ received funding from the German Federal Environmental Foundation (Deutsche Bundesstiftung Umwelt). Fieldwork was supported by NSF grants (1806213, 2336164) and AWI expedition baseline funding.

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
