# Peer review of "Sedimentary insights into organic matter alteration in Arctic Alaska's saline permafrost"

_EGUsphere, 2025_

## Referee Comment (RC1)

Reviewer comment – EGUsphere Open Review

Manuscript: Sedimentary insights into organic matter alteration in Arctic Alaska's saline permafrost

**General assessment**

This manuscript presents valuable data on organic matter (OM) alteration processes in saline permafrost sediments from Arctic Alaska. The topic is highly relevant to understanding carbon cycling and permafrost-climate feedbacks in rapidly warming Arctic systems. However, the current version would benefit from clarification of sampling strategy, clearer data grouping logic, and an expanded discussion on the broader implications of saline permafrost thaw. Figures and tables could also be improved for interpretability.

Overall, the study has great potential, but the manuscript would be strengthened by addressing the following minor points.

**Comments**

(Lines 35–45)

Expand the discussion of how increased salinity can enhance ground warming, potentially accelerating carbon degradation. Clarifying this physical–biogeochemical linkage would strengthen the introduction.

(Section 2.21, Line 145)

It is not clear at this part whether only surface sediments or deeper permafrost cores were analysed. This is important, as surface layers are likely subject to strong mixing due to seasonal freeze–thaw dynamics.Please provide sampling intervals, total number of samples per core, and core depths. Without this, the representativeness of the dataset cannot be evaluated. Consider whether surface scouring or removal by groundfast ice might have influenced surface sediment preservation, as this could bias near-surface data (see Table S1).

Figure 1: Consider enlarging or dividing into two panels (e.g., adding a panel showing core depths) to improve readability and convey stratigraphic context. The current layout is difficult to read. Either enlarge the figure or add a second panel showing core depths to improve interpretability.

Table 3: The rationale for grouping samples after individual core analyses remains unclear. Please clarify the purpose and implications of this grouping, particularly why some groups were excluded from the PERMANOVA due to small sample sizes (n < 5). The statistical framework appears rigorous and well executed, but it becomes somewhat detached from the broader scientific narrative. Consider linking these statistical groupings more clearly to the ecological and biogeochemical processes

discussed elsewhere in the paper to ensure that the results contribute directly to understanding the mechanisms of organic matter alteration under different thermal and saline regimes.

(Lines ~535–580)

The discussion could be expanded to better situate the findings within the broader permafrost–climate context. At present, there is no information provided on lake surface areas or volumes, which would help to contextualize how these systems influence thermal forcing and potential greenhouse gas emissions. Including even basic estimates or references for lake morphology would strengthen the discussion. Moreover, while the manuscript effectively documents organic matter alteration across thermal and saline regimes, it stops short of linking these findings to methane production or carbon loss processes in saline permafrost environments. A qualitative discussion of how organic matter degradation under saline, unfrozen conditions may contribute to methane or $CO_2$ release would significantly enhance the manuscript's broader relevance. In addition, the term "availability" (around line 570) could be replaced with a more precise descriptor such as "degradation potential" or "reactivity" to better reflect the geochemical processes described. Finally, a clearer statement on why saline permafrost thaw is globally significant would be valuable—ideally supported by literature estimates of the spatial extent and carbon pool size of saline permafrost deposits. These additions would help connect the strong sedimentary and statistical analyses to the larger climatic implications of thawing saline permafrost.

**Overall recommendation**
This study presents a strong and well-structured dataset with great potential to advance understanding of organic matter alteration in saline permafrost systems. The manuscript would benefit from greater methodological transparency and clearer contextual framing to fully highlight its scientific importance. I recommend a minor revision focusing on the following key aspects:

Clarify the sampling design and rationale for data grouping, particularly in Table 3, to ensure the statistical analyses are clearly linked to the environmental processes being studied.

Improve figure readability and presentation (especially Figure 1) to better convey stratigraphic and spatial relationships.

Expand the discussion to more explicitly address the broader climatic implications of saline permafrost thaw, including its potential influence on methane production, carbon degradation, and Arctic greenhouse gas feedbacks.

Addressing these points would substantially improve the manuscript's clarity, coherence, and overall impact within the context of Arctic carbon cycle research.

---

## Referee Comment (RC2)

Review of manuscript: Sedimentary insights into organic matter alteration in Arctic Alaska's saline Permafrost.

**General comments**

The manuscript of Seeman and co-authors present organic matter characteristic (TOC, alkane and alkane-derived proxies) in deposits along a salinity gradient including permafrost and active layers in Arctic Alaska. They target a unique type of deposit, rarely studied so this wealth of data is definitely interesting to publish. However, I would recommend presenting the proxies used in this study a bit better, especially their limitations.

Generally the discussion is hard to follow and the manuscript would benefit from a bit more organization in this part, maybe grouping the type of observation together rather than for each type of deposit? Or maybe focusing more on the effect of salinity on the observed degradation?

**Specific comments**

L39-40: "Therrmokarst processes are accelerating in the Alaskan tundra (Chen et al., 2021) while Nitze et al. (2017) describe a thermokarst lake drainage trend." I'm not sure I clearly understand this sentence, especially the use of "while".

L42: can you explain what is "saline permafrost"?

L45: same for "unfrozen cryotic conditions"

L58: could you give a range of expected salinities

L59: what do you mean with "differentiated"? More detailed, or with different techniques? It would be good to refer to the techniques used before in Giest et al. 2025 and expend on what will be newly applied in this study.

L62-64: I agree with the authors that CPI has been increasingly used but it has major bias, in particular in region with old rock deposit that can lower the CPI. It would be good to present the organic carbon proxies and their limitation to be clear with the readers. Similarly d13C, D14C, C:N ratios have bias that need to be presented (heterogenous source effect, post deposition transformation, …).

L77-78: is the temperature average from a meteorological station? IF so which one and how close to the study area is it?

L83-86: I'm not sure this part on vegetation in the region is needed as all the sites are mainly aquatic

L90: 22 and 50% of what is covered by thermokarst and DLB?

L154: for this part the subsamples were freeze dried?

L159: Since the samples were not acidified TOC determined with a SoliTOC can be overestimated as some carbonate already burn before 900C. This is not an issue but should be acknowledged.

L171-172: "Stable carbon isotope ratios are commonly applied as a proxy for organic matter origin and degradation in permafrost regions (e.g., Alewell et al., 2011; Strauss et al., 2015).", as in the paragraph before this technical explanation should come after the method explanation.

L170-176: Can you give the standards used for this analysis as well as the measurement error.

L180: How was the radiocarbon dating conducted, what pre-treatments were done on the samples? Were the bulk sediment samples acidified?

L185: "eluted" or "extracted"

L190: how was the medium pressure liquid chromatography performed, with which solvents?

L201-202: "transformation effect", do you mean degradation?

L211-212: another limitation of the ACL and n-alkane proxy is the heterogeneity and potential overlap of the source, see the review of Diefendorf et al., 2011

L214: There is an odd over even predominance in terrestrial vegetation. In hypersaline environment the contrary can be observed (e.g. Li et al., 2024 Salinity impacts on *n*-alkanes in lake sediments of the Badain Jaran Desert, Northwestern China: Implications for paleoclimate reconstruction; Samantaray and Sanyal 2023 Effect of salinity on the preservation of plant-derived *n*-alkyl compounds in the terrestrial-aquatic interface). This effect of salinity might be relevant for the study site.

L259-264: Since TOC varies so much between units (35 to 5%) it would be more informative to express concentration normalized by TOC (ng/gOC) so that differences between units actually reflect different alkane concentration and not just the TOC effect.

Figure 3: I don't see any red point in the figure, which incubation is referenced in the caption? It is not described in the method. I think the 14C ages should be added next to the depth to give a better idea of the period captured by the cores.

Paragraph 4.1. This paragraph has a lot of results instead of discussion and can be shortened by moving the core unit description into the results section. The interpretation of the different thaw process and organic matter input fits well in the discussion.

L397-399: There is no explanation of the claim that ACL values support a shift from grass to a more mixed vegetation in the early Holocene.

L399-403: Paq limitation is presented but just brushed aside without any reason (how is Betula shrub input influencing Paq for example?). In general this whole paragraph investigation the changes in ACL and Paq is not well described and there is no clear support in the text or in the figures.

L413-414: What is the consequence of finding brackish talik sediment in a lake that was previously described as fresh? I get the point but this is not clearly explained. Also when did East Twin Lake experiences a transition to brackish water?

L417: Is the Teshepuk lake area far from the studied sites?

L417-418: Could Paq also indicate increased input from Betula as mentioned in the paragraph before? Which would fit with the info from ACL and n-alkane ratio?

L419-423 "At this point, it needs to be stressed that in our study the ACL decreases with decreasing CPI values (r = 0.79, p < 0.001), meaning that the vegetation signal is influenced by organic matter degradation (strongest in ETL). This is a commonly observed process (e.g., Jongejans et al., 2020; Struck et al., 2018), which needs consideration when interpreting *n*-alkane proxies." This statement is coming a bit late and can be presented in the results already or at the beginning of the discussion. Why are the authors still using it if the main control on ACL is OM degradation?

L432: Can you indicate again what material has been dated for this site? The age difference could be due to the type of material.

L452-453: Would it be better to compare your lagoon with north American lagoon TOC and TN data like those in the Tuktoyaktuk area?

L466-467: Can you give some numbers? In general in part 4.2. it would help the reader to get some numbers, averages …

L469-470: Can you give a reference for " $\delta$ 13C values become lighter (less negative) with degradation"

L479-480: I don't think there is much of a difference between 6.5 and 6.9 for a CPI value. If you think this is a significant difference, can you cite similar setting where such a small difference has been interpreted.

L495: please give standard deviation and number of point when you give an average for transparency.

**Technical corrections**

Throughout the text: There are some space missing before references, likely dues to reference formatting (for example L75).

Throughout the text: avoid the formulation "we" and use passive sentence throughout the text

L31: "the polar north" could instead be "the poles"

L33: "which relate […] to climate change" maybe to be more precise write to "temperature change"

L37: I think what matters most here is that these plains are low elevation? Instead of "vast"?

L42: "Furthermore" is maybe not needed here as there is a new paragraph starting

L48 "coast" instead of "coastlines"

L608-609: The font differs for the last sentences

---

## Author Comment (AC1)

Reviewer comment – EGUsphere Open Review

Manuscript: Sedimentary insights into organic matter alteration in Arctic Alaska's saline permafrost

**General assessment**

This manuscript presents valuable data on organic matter (OM) alteration processes in saline permafrost sediments from Arctic Alaska. The topic is highly relevant to understanding carbon cycling and permafrost-climate feedbacks in rapidly warming Arctic systems. However, the current version would benefit from clarification of sampling strategy, clearer data grouping logic, and an expanded discussion on the broader implications of saline permafrost thaw. Figures and tables could also be improved for interpretability.

Author's response (AR): Thank you for your time spending reviewing our manuscript and all your valuable feedback. These will substantially improve the manuscript. Please find below our specific replies to your comments.

Overall, the study has great potential, but the manuscript would be strengthened by addressing the following minor points.

**Comments**

(Lines 35–45)

Expand the discussion of how increased salinity can enhance ground warming, potentially accelerating carbon degradation. Clarifying this physical–biogeochemical linkage would strengthen the introduction.

AR: Thank you for pointing out, that this linkage should be explained early on in the paper. The paragraph has now been extended to demonstrate the critical role of salts in permafrost carbon dynamics (line 42-50).

(Section 2.21, Line 145)

It is not clear at this part whether only surface sediments or deeper permafrost cores were analysed. This is important, as surface layers are likely subject to strong mixing due to seasonal freeze–thaw dynamics. Please provide sampling intervals, total number of samples per core, and core depths. Without this, the representativeness of the dataset cannot be evaluated. Consider whether surface scouring or removal by groundfast ice might have influenced surface sediment preservation, as this could bias near-surface data (see Table S1).

AR: Thank you for stressing these points. We changed this accordingly: In the method sections 2.2.1 and 2.2.2 details are added now. Fieldwork aimed at sampling near-surface and deep (> 1 m) sediments, which was reached with the exception of coring at West Twin Lake (0.23 m) and the semi-open lagoon (0.46 m). The sampling interval was roughly 5 cm. Total sample numbers were added to Table S1, which also lists core depths. With the exception of East Twin Lake, no artificial disturbances of the cores were observed.

Figure 1: Consider enlarging or dividing into two panels (e.g., adding a panel showing core depths) to improve readability and convey stratigraphic context. The current layout is difficult to read. Either enlarge the figure or add a second panel showing core depths to improve interpretability.

AR: Indeed, this is a large figure and it needs to be printed large on the page in the final manuscript. We hope that this will happen with copy-editing the paper in the final stage. We kindly disagree with dividing the figure into 2 panels, as this visualization in one panel is the only way to compare parameters and individual cores as easy as possible. The core depths can be read on the y-axes and the units were added to improve readability. The figure will be slightly edited for better readability and the caption has now been adjusted by adding the maximum core depth for each core.

Table 3: The rationale for grouping samples after individual core analyses remains unclear. Please clarify the purpose and implications of this grouping, particularly why some groups were excluded from the PERMANOVA due to small sample sizes (n < 5). The statistical framework appears rigorous and well executed, but it becomes somewhat detached from the broader scientific narrative. Consider linking these statistical groupings more clearly to the ecological and biogeochemical processes discussed elsewhere in the paper to ensure that the results contribute directly to understanding the mechanisms of organic matter alteration under different thermal and saline regimes.

AR: Thank you very much. As we think thermal and salinity stages are worth being considered individually concerning their influence on organic carbon characteristics, the combined impact is also the most closes to natural reality, and thus valuable to investigate. Especially, as both categories relate to another (i.e., unfrozen sediments tend to have higher salinities). You are absolutely right; the focus of the paper may appear a bit diluted with these detailed analyses. However, the combination of both factors is important as seen by the statistical results which revealed that statistically significant differences exist between groups. These results are therefore also further discussed in sections 4.1.3 and 4.2. For these reasons we would like to keep the combined PERMANOVA in the manuscript.
Including your valuable feedback, we did the following revision: In section 3.2.3 an argumentation on the importance of this analyses is included now and Table 3 has been adjusted for better readability. If any group had low sample counts (n < 5), these were generally excluded from the statistics for valid representativeness. This is now stressed in section 3.2.3.

(Lines ~535–580)

The discussion could be expanded to better situate the findings within the broader permafrost–climate context. At present, there is no information provided on lake surface areas or volumes, which would help to contextualize how these systems influence thermal forcing and potential greenhouse gas emissions. Including even basic estimates or references for lake morphology would strengthen the discussion.

AR: Thank you for these valuable comments. The specific impacts on the waterbodies and for the study region will be further discussed. For that, chosen lake morphometrics will be provided in section 2.1 (Study area). Further information can be found in Table S1. The lakes cover ca. 1.3 $km^2$ each and are about 2 m deep, which lies within the range of typical lakes in the region (Arp et al., 2011). Their impact on landscape evolution is great, since a trend from bedfast to floating ice conditions, such as observed for East Twin Lake (Jones et al., 2023) amplifies permafrost

(carbon) degradation. We now mention the role of thermokarst lakes more explicit in line 574-576.

Moreover, while the manuscript effectively documents organic matter alteration across thermal and saline regimes, it stops short of linking these findings to methane production or carbon loss processes in saline permafrost environments. A qualitative discussion of how organic matter degradation under saline, unfrozen conditions may contribute to methane or $CO_2$ release would significantly enhance the manuscript's broader relevance.

AR: This is a very important point, you are right. We included potential greenhouse gas production thin in the submitted version already (lines 555-599), but it seemed to be too hidden. Therefore, we now try to put more spotlight on this point by adjusting the paragraph in line 578-582: "From a microbiological point of view, this can be explained by microbial communities that are likely dominated by methanogens ($CH_4$ producers) in West Twin Lake while in East Twin Lake sulfate-reducing microbes ($CO_2$ producers) probably established additionally due to the saline deposits. A potential co-existence of these microbial communities may lead to enhanced greenhouse gas production (Jenrich et al., 2024; Yang et al., 2023)." Specific conditions and processes however cannot be assessed in this paper and is subject to future research.

In addition, the term "availability" (around line 570) could be replaced with a more precise descriptor such as "degradation potential" or "reactivity" to better reflect the geochemical processes described.

AR: We changed the term "availability to "reactivity".

Finally, a clearer statement on why saline permafrost thaw is globally significant would be valuable—ideally supported by literature estimates of the spatial extent and carbon pool size of saline permafrost deposits. These additions would help connect the strong sedimentary and statistical analyses to the larger climatic implications of thawing saline permafrost.

AR: The distribution and (carbon) properties of saline permafrost is generally not well studied. Our study contributes at filling this gap, but large-scale studies are currently lacking. We added this discussion point now in line 602-605: "Overall, observed carbon and landscape dynamics can be expected to play a major role across the wider saline permafrost region. Brouchkov (2003) published an estimated map of the saline permafrost zone, however large-scale quantifications of the saline permafrost distribution and its properties, including organic carbon stocks, are currently lacking. This calls for future study efforts, which would constitute important steps forward in understanding the role of salt in Arctic permafrost."

**Overall recommendation**
This study presents a strong and well-structured dataset with great potential to advance understanding of organic matter alteration in saline permafrost systems. The manuscript would benefit from greater methodological transparency and clearer contextual framing to fully highlight its scientific importance. I recommend a minor revision focusing on the following key aspects:

AR: Thank you very much for your support.

Clarify the sampling design and rationale for data grouping, particularly in Table 3, to ensure the statistical analyses are clearly linked to the environmental processes being studied.

AR: Changed accordingly. We are happy to send a revised version with track changes as soon as possible.

Improve figure readability and presentation (especially Figure 1) to better convey stratigraphic and spatial relationships.

AR: Please see our comment from above. Enlarging the figure in the final manuscript with a revised caption explaining e.g. the core depth more would be our preferred revision.

Expand the discussion to more explicitly address the broader climatic implications of saline permafrost thaw, including its potential influence on methane production, carbon degradation, and Arctic greenhouse gas feedbacks.

AR: Changes accordingly applied in the section 4.3 of the revised document.

Addressing these points would substantially improve the manuscript's clarity, coherence, and overall impact within the context of Arctic carbon cycle research.

AR: Thank you very much for your very constructive feedback.

---

## Author Comment (AC2)

Review of manuscript: Sedimentary insights into organic matter alteration in Arctic Alaska's saline Permafrost.

**General comments**

The manuscript of Seeman and co-authors present organic matter characteristic (TOC, alkane and alkane-derived proxies) in deposits along a salinity gradient including permafrost and active layers in Arctic Alaska. They target a unique type of deposit, rarely studied so this wealth of data is definitely interesting to publish. However, I would recommend presenting the proxies used in this study a bit better, especially their limitations.

Author's response (AR): Thank you very much for your positive and constructive feedback and support.

Generally the discussion is hard to follow and the manuscript would benefit from a bit more organization in this part, maybe grouping the type of observation together rather than for each type of deposit? Or maybe focusing more on the effect of salinity on the observed degradation?

AR: Thank you very much. Basing on your comment we now introduce for better readability in the first discussion section (4.1) subsections. Concerning the regrouping of the parameter we kindly disagree and hope to convince you that only grouping after landscape position enables the clear interpretation of degradation with landscape evolution.
Because this study applies a multiproxy approach, discussing each parameter individually would overemphasize methodology and blur the integrated landscape interpretation and joint changes that are key to our study.

**Specific comments**

L39-40: "Therrmokarst processes are accelerating in the Alaskan tundra (Chen et al., 2021) while Nitze et al. (2017) describe a thermokarst lake drainage trend." I'm not sure I clearly understand this sentence, especially the use of "while".

AR: Thank you for pointing this out, indeed the sentence was formulated unclear. It is now reformulated to: "Thermokarst processes are accelerating in the Alaskan tundra (Chen et al., 2021), and Nitze et al. (2017) describe a concurrent trend of thermokarst lake drainage." (line 38-39).

L42: can you explain what is "saline permafrost"?

AR: Thank you for indicating that this paragraph needs clarification. It has now been adjusted (line 42-50). Saline permafrost refers to perennially frozen sediments which contain salts. These sediments are vulnerable to thaw, as the thawing point is depressed in these deposits.

L45: same for "unfrozen cryotic conditions"

AR: The paragraph has been adjusted as mentioned before. Unfrozen cryotic conditions refer sediments which are unfrozen although temperatures are below the freezing point.

L58: could you give a range of expected salinities

AR: The salinities observed in coastal permafrost environments range between freshwater and hypersaline conditions. This information is now added (line 61-63): "Since coastal permafrost regions can vary substantially in their salinity - between fresh and hypersaline porewater conditions - (Jenrich et al., 2021) , more differentiated investigations concerning different salinity levels are needed.".

L59: what do you mean with "differentiated"? More detailed, or with different techniques? It would be good to refer to the techniques used before in Giest et al. 2025 and expend on what will be newly applied in this study.

AR: Thank you for pointing out this unclarity. The sentence is now re-formulated. Giest et al., (2025) only differentiated between saline and non-saline sediments. Since salinities may vary substantially, however, different salinity levels should also be differentiated from another. This is the new approach implemented here.

L62-64: I agree with the authors that CPI has been increasingly used but it has major bias, in particular in region with old rock deposit that can lower the CPI. It would be good to present the organic carbon proxies and their limitation to be clear with the readers. Similarly d13C, D14C, C:N ratios have bias that need to be presented (heterogenous source effect, post deposition transformation, …).

AR: Thank you for raising this important point. We fully acknowledge that each individual proxy used in this study has inherent limitations and potential biases, including those related to source heterogeneity and post-depositional alteration.
This consideration was a key motivation for applying an extensive multiproxy approach, including n-alkane biomarkers, rather than relying on single parameters. In addition, our statistical analyses are not based on individual proxies alone; instead, we apply a multivariate PERMANOVA framework that integrates the full proxy suite, thereby reducing the influence of biases associated with any single parameter.
To address the reviewer's concern more explicitly, we have now added a brief overview of general proxy limitations in Section 2.2.2. Proxy-specific limitations that are particularly relevant for our dataset (e.g. alteration of ACL values due to carbon degradation) are discussed in detail in the discussion section (see response below).

L77-78: is the temperature average from a meteorological station? IF so which one and how close to the study area is it?

AR: Yes, the data presented originates from the meteorological station in Utqiaġvik. Details of the data can be found in Rawlins (2021), as cited accordingly. The distance between the town and the study area is about 10 km (line 78).

L83-86: I'm not sure this part on vegetation in the region is needed as all the sites are mainly aquatic

AR: The upland and the DLB have mainly terrestrial vegetation. Some basic vegetation information is crucial as a background for the alkane proxies as this topic is picked up in the discussion (section 4.1). According to your comment we see shortening potential, thus the information provided is now slightly reduced.

L90: 22 and 50% of what is covered by thermokarst and DLB?

AR: The sentence is now reformulated: "Thermokarst lakes cover about 22 % and DLBs 50 % of the landscape (Hinkel et al., 2003; Jones et al., 2022)."

L154: for this part the subsamples were freeze dried?

AR: Yes exactly. As now adjusted in the sentence before, samples were freeze-dried after porewater extraction. Subsequent sample treatment followed with freeze-dried samples.

L159: Since the samples were not acidified TOC determined with a SoliTOC can be overestimated as some carbonate already burn before 900C. This is not an issue but should be acknowledged.

AR: Thank you for bringing this up, we are also aware of this. For marine sediments or samples with a substantial inorganic carbon content we would precede with sample acidification before quantification. However, as mentioned, for our samples this is not an issue and is therefore not mentioned in the manuscript.

L171-172: "Stable carbon isotope ratios are commonly applied as a proxy for organic matter origin and degradation in permafrost regions (e.g., Alewell et al., 2011; Strauss et al., 2015).", as in the paragraph before this technical explanation should come after the method explanation.

AR: Thank you for pointing out this inconsistence. The paragraph is now adjusted accordingly.

L170-176: Can you give the standards used for this analysis as well as the measurement error.

AR: The $\delta^{13}$C measurements were run at the AWI ISOLAB facility which uses lab specific standards (external standards) according to the procedure described in Schwamborn et al 2022 : "External standards were used to control the instrument precision and the range of replicate stable carbon isotope measurements was generally less than 0.15‰.". We now cited this reference, but would like to keep the paper concise and therefore refer to the reference here.

L180: How was the radiocarbon dating conducted, what pre-treatments were done on the samples? Were the bulk sediment samples acidified?

AR: Detailed laboratory procedures can be found in Mollenhauer et al. (2021) as cited in the manuscript. We would like to keep the paper concise and therefore refer to the reference here.

L185: "eluted" or "extracted"

AR: Thanks, the term has been changed to "extracted".

L190: how was the medium pressure liquid chromatography performed, with which solvents?

AR: Medium pressure chromatography uses n-hexane over silica columns. This information was added to the paragraph.

L201-202: "transformation effect", do you mean degradation?

AR: Indeed. The wording has now been adjusted.

L211-212: another limitation of the ACL and n-alkane proxy is the heterogeneity and potential overlap of the source, see the review of Diefendorf et al., 2011

AR: Thank you for stressing this point. The limitations are now mentioned specifically: ACL limitations concern the blindness towards gymnosperms, overlapping chemotaxonomic patterns of different source material and potential post depositional alteration through degradation (Diefendorf et al., 2011; Jongejans et al., 2020; Struck et al., 2018; Zech et al., 2021).

L214: There is an odd over even predominance in terrestrial vegetation. In hypersaline environment the contrary can be observed (e.g. Li et al., 2024 Salinity impacts on *n*-alkanes in lake sediments of the Badain Jaran Desert, Northwestern China: Implications for paleoclimate reconstruction; Samantaray and Sanyal 2023 Effect of salinity on the preservation of plant-derived *n*-alkyl

compounds in the terrestrial-aquatic interface). This effect of salinity might be relevant for the study site.

AR: Thank you for pointing towards these references. Li et al. (2024) found that salinity enhanced alkane degradation. An odd over even predominance still occurs, as found in our study. The fact that salinity itself might already accelerate carbon degradation is now included in the discussion (section 4.2, line 517-518).

L259-264: Since TOC varies so much between units (35 to 5%) it would be more informative to express concentration normalized by TOC (ng/gOC) so that differences between units actually reflect different alkane concentration and not just the TOC effect.

AR: Thank you for this suggestion. The study aims at providing absolute/in-situ numbers, which is the total alkane content per gram sediment. The relative distribution of TAC contents among the samples does not change significantly if normalized with TOC contents (see Pangaea data source: https://doi.pangaea.de/10.1594/PANGAEA.983966). However, TACs normalized to TOC contents are now reported in the results additionally.

Figure 3: I don't see any red point in the figure, which incubation is referenced in the caption? It is not described in the method. I think the 14C ages should be added next to the depth to give a better idea of the period captured by the cores.

AR: Thank you for pointing this out. This is now corrected. Since radiocarbon ages are already presented in Figure 2, we agreed to not repeat those data points in Figure 3.

Paragraph 4.1. This paragraph has a lot of results instead of discussion and can be shortened by moving the core unit description into the results section. The interpretation of the different thaw process and organic matter input fits well in the discussion.

AR: This section has now been slightly shortend. We think that core units are beneficial for the discussion, as it is well possible to lead the reader through the argumentations with these.

L397-399: There is no explanation of the claim that ACL values support a shift from grass to a more mixed vegetation in the early Holocene.

AR: Thank you for noticing. The sentence is now reformulated: "Considering the ACL and *n*-alkane ratio, it could be inferred that this slowly occurring shrubification throughout the Holocene can be confirmed by our *n*-alkane analysis due to the tendency towards lower ACL and *n*-alkane ratio values in Holocene samples compared to the deepest (Lateglacial) biomarker sample (Fig. 3a)". Expected value ranges for different vegetation types can be found in the method section.

L399-403: Paq limitation is presented but just brushed aside without any reason (how is Betula shrub input influencing Paq for example?). In general this whole paragraph investigation the changes in ACL and Paq is not well described and there is no clear support in the text or in the figures.

AR: Thanks for pointing out, we adjusted this accordingly. Betula shrubs limit the power of Paq since these are reported with mid-chain lengths just as aquatic vegetation (see Weber and Schwark, 2020 and methods section) and the chain lengths are used to calculate Paq. This means, that Paq might also pick up Betula shrub signals. Together with the clarifications of your last point, the paragraph is now improved.

L413-414: What is the consequence of finding brackish talik sediment in a lake that was previously described as fresh? I get the point but this is not clearly explained. Also when did East Twin Lake experiences a transition to brackish water?

AR: The consequence of brackish sediment in the freshwater lake is, that the lake will likely turn brackish with continued subsidence. In East Twin Lake sediments an increase of electrical conductivity by about 200 % occurred between 2016 and 2022 (Jones et al., 2023), highlighting the temporal dynamic of this process. The paragraph has now been adjusted accordingly.

L417: Is the Teshepuk lake area far from the studied sites?

AR: It lies about 120 km east of our study area. This information is now included.

L417-418: Could Paq also indicate increased input from Betula as mentioned in the paragraph before? Which would fit with the info from ACL and n-alkane ratio?

AR: Yes, Betula might play a role here, too. Potential terrestrial vegetation input is mentioned subsequently by arguing that the ACL and *n*-alkane ratio provide evidence for this.

L419-423 "At this point, it needs to be stressed that in our study the ACL decreases with decreasing CPI values (r = 0.79, p < 0.001), meaning that the vegetation signal is influenced by organic matter degradation (strongest in ETL). This is a commonly observed process (e.g., Jongejans et al., 2020; Struck et al., 2018), which needs consideration when interpreting *n*-alkane proxies." This statement is coming a bit late and can be presented in the results already or at the beginning of the discussion. Why are the authors still using it if the main control on ACL is OM degradation?

AR: Thank you for pointing this out. The limitation of the ACL is now added in the methods section. It is mentioned again at this point because degradation is strongest in East Twin Lake. The ACL is still a valuable tool for vegetation reconstructions, since it is an alternative/addition to classic paleo proxies such as C:N and d13C.

L432: Can you indicate again what material has been dated for this site? The age difference could be due to the type of material.

AR: In our study (Table S8), as well as in Brown et al. (2003) plant remains were dated to determine peat sequestration onset. Therefore, the age difference cannot be explained by type of material.

L452-453: Would it be better to compare your lagoon with north American lagoon TOC and TN data like those in the Tuktoyaktuk area?

AR: Yes, we additionally included now the Teshepuk Lake area (Giest et al., 2025) and the Mackenzie Delta region (Jenrich et al., 2025).

L466-467: Can you give some numbers? In general in part 4.2. it would help the reader to get some numbers, averages …

AR: Thank you for this suggestion. Numbers have been added to the section, but we also tried to keep the section tight without repeating too many results.

L469-470: Can you give a reference for " $\delta$ 13C values become lighter (less negative) with degradation"

AR: Yes, Strauss et al. (2015) is cited in the sentence.

L479-480: I don't think there is much of a difference between 6.5 and 6.9 for a CPI value. If you think this is a significant difference, can you cite similar setting where such a small difference has been interpreted.

AR: We would not state that this is a significant difference but we think it might be important to point out this difference as degradation under unfrozen cryotic and saline conditions may be relevant here. At this point, we therefore also cite now Li et al. (2024), which is the publication you suggested earlier.

L495: please give standard deviation and number of point when you give an average for transparency.

AR: Standard deviations are now added to numbers.

**Technical corrections**

Throughout the text: There are some space missing before references, likely dues to reference formatting (for example L75).

AR: Thank you for pointing this out. Missing spaces have been added now.

Throughout the text: avoid the formulation "we" and use passive sentence throughout the text

AR: Thank you for this comment. We agree that passive constructions are commonly used in methodological descriptions. We therefore adjusted selected sentences in the Methods section, while retaining first-person plural elsewhere to ensure clarity and consistency. A full conversion to passive voice throughout the manuscript was not undertaken.

L31: "the polar north" could instead be "the poles"

AR: It should be "polar north" in this case, as this sentence refers to the Arctic.

L33: "which relate […] to climate change" maybe to be more precise write to "temperature change"

AR: The wording has been changed to "global warming".

L37: I think what matters most here is that these plains are low elevation? Instead of "vast"?

AR: Yes coastal plains are of low elevation, but the large extent is also relevant to mention to stress the importance of these regions. Therefore, the wording is kept as is.

L42: "Furthermore" is maybe not needed here as there is a new paragraph starting

AR: "Furthermore" is now removed from the sentence.

L48 "coast" instead of "coastlines"

AR: Plural is required here, since the processes generally affect Arctic coasts. Thus, the sentence is kept in its current form.

L608-609: The font differs for the last sentences

AR: The text is correctly formatted now.

AR: The text is correctly formatted now.